# Vitamin B2 enables regulation of fasting glucose availability

Peter M Masschelin[1,2,3], Pradip Saha[1,2], Scott A Ochsner[3], Aaron R Cox[1,2], Kang Ho Kim[4], Jessica B Felix[1,2,3], Robert Sharp[1,2], Xin Li[1,2], Lin Tan[5], Jun Hyoung Park[6], Liping Wang[7], Vasanta Putluri[3], Philip L Lorenzi[5], Alli M Nuotio-Antar[8], Zheng Sun[1,2], Benny Abraham Kaipparettu[6], Nagireddy Putluri[3], David D Moore[3,9], Scott A Summers[7], Neil J McKenna[3], Sean M Hartig[1,2,3]*

[1]Department of Diabetes, Endocrinology, and Metabolism, Baylor College of Medicine, Houston, United States; [2]Department of Medicine, Baylor College of Medicine, Houston, United States; [3]Department of Molecular and Cellular Biology, Baylor College of Medicine, Houston, United States; [4]Department of Anesthesiology, University of Texas Health Sciences Center, Houston, United States; [5]Department of Bioinformatics and Computational Biology, The University of Texas MD Anderson Cancer Center, Houston, United States; [6]Department of Molecular and Human Genetics, Baylor College of Medicine, Houston, United States; [7]Department of Nutrition and Integrative Physiology, University of Utah, Salt Lake City, United States; [8]Department of Pediatrics, Baylor College of Medicine, Houston, United States; [9]Department of Nutritional Sciences and Toxicology, University of California, Berkeley, Berkeley, United States

**\*For correspondence:**
hartig@bcm.edu

**Competing interest:** The authors declare that no competing interests exist.

**Abstract** Flavin adenine dinucleotide (FAD) interacts with flavoproteins to mediate oxidation-reduction reactions required for cellular energy demands. Not surprisingly, mutations that alter FAD binding to flavoproteins cause rare inborn errors of metabolism (IEMs) that disrupt liver function and render fasting intolerance, hepatic steatosis, and lipodystrophy. In our study, depleting FAD pools in mice with a vitamin B2-deficient diet (B2D) caused phenotypes associated with organic acidemias and other IEMs, including reduced body weight, hypoglycemia, and fatty liver disease. Integrated discovery approaches revealed B2D tempered fasting activation of target genes for the nuclear receptor PPARα, including those required for gluconeogenesis. We also found PPARα knockdown in the liver recapitulated B2D effects on glucose excursion and fatty liver disease in mice. Finally, treatment with the PPARα agonist fenofibrate activated the integrated stress response and refilled amino acid substrates to rescue fasting glucose availability and overcome B2D phenotypes. These findings identify metabolic responses to FAD availability and nominate strategies for the management of organic acidemias and other rare IEMs.

## Editor's evaluation

This paper provides important findings on the metabolic roles of vitamin B2 in the liver that will be of broad interest. Convincing data establish the effects of vitamin B2 deficiency on body composition, energy expenditure, and glucose metabolism.

## Introduction

Flavin mononucleotide (FMN) and flavin adenine dinucleotide (FAD) serve as essential cofactors for diverse proteins that mediate oxidation-reduction (redox) reactions, transcriptional regulation, and metabolism (*Powers, 2003*). In particular, FAD supports the activity of flavoproteins that enable the electron transport chain (ETC), the tricarboxylic acid (TCA) cycle, and fatty acid oxidation (FAO). Along these lines, mutations occurring in more than 50% of human flavoproteins cause inborn errors of metabolism (IEMs) with heterogeneous clinical presentations frequently characterized by organic acidemia, fasting intolerance, and fatty liver disease (*Balasubramaniam et al., 2019*). Compared to the more well-understood roles of nicotinamide adenine dinucleotide (NAD), the physiological relevance of FAD has remained largely ignored. This gap in knowledge slows the pursuit of therapeutic strategies to treat IEM and leaves fundamental energy balance roles for FAD unresolved.

The liver coordinates whole-body metabolism during fasting by releasing glucose and other fuels to spare the brain and survive low-nutrient conditions. Nuclear receptors sense and receive the signals of nutrient abundance in the liver to perform precise regulation of genes that ultimately maintain the energetic needs of the FAO and amino acid catabolism pathways serving gluconeogenesis (*Scholtes and Giguère, 2022*). Among nuclear receptors, peroxisome proliferator-activated receptor a (PPARα) modulates fasting responses and pivots hepatocytes toward conservation and recycled substrates to sustain energy production. NAD can activate nuclear receptor co-activators in the liver for gluconeogenesis (*Rodgers et al., 2005*). However, the role of FAD as a bridging molecule for metabolic reactions and transcriptional regulation of energy balance are almost completely unknown. Synthetic PPARα agonists promote FAO by directing the activity of pathways for balanced lipid metabolism in the liver, which consequently supported a series of FDA-approved fibrate drugs for the treatment of hypertriglyceridemia (*Jackevicius et al., 2011*). The lipid-lowering properties of bezafibrate and fenofibrate motivated preclinical studies (*Steele et al., 2020*; *Waskowicz et al., 2019*; *Yavarow et al., 2020*) that form ongoing efforts to overcome the limited armament of therapies for IEMs.

Nutrition impacts FAD availability for the organism and dietary riboflavin (vitamin B2) supplies the backbone for all FAD and FMN synthesis. Riboflavin deficiency gives rise to abnormal development and energy balance disorders. Low riboflavin also occurs in more broadly observed diseases including cancer (*Kabat et al., 2008*; *Powers, 2005*), HIV (*Beach et al., 1992*; *Fouty et al., 1998*), and cardiovascular disease (*Tzoulaki et al., 2012*). Thus, models that expand how FAD requirements form the regulatory environment for metabolic homeostasis provide opportunities for preclinical experiments and studies of fundamental nutrient-sensing and disease mechanisms. Here, we define the outcomes of vitamin B2 depletion resembling IEMs of flavoprotein deficiency and transcriptional pathways that preserve glucose availability in fasted mice.

## Results

### Riboflavin deficiency alters body composition and energy expenditure

In mammals, diet furnishes vitamin B2 to synthesize all the FAD for electron transfer in the mitochondria and redox reactions required for cellular homeostasis (*Powers, 2003*). Among key metabolic organs, ad libitum FAD levels were highest in the liver, heart, and kidney (*Figure 1a*). To study how FAD depletion influences energy balance, we exposed male mice to vitamin B2-deficient (B2D) or control diets for 4 wk and performed metabolic phenotyping (*Figure 1b*). We found that 99% B2D was sufficient to reduce liver FAD levels by 70% (*Figure 1c*). Moreover, B2D significantly blunted weight gain (*Figure 1d and e*), and body composition measurements showed B2D also reduced fat mass (*Figure 1f*). When we examined the contribution of B2 to energy expenditure, we were surprised the stunted body weight phenotype of B2D did not arise from higher oxygen consumption or canonical markers of brown adipose tissue thermogenesis (*Figure 1—figure supplement 1*). B2D strongly reduced oxygen consumption (*Figure 1g*), but substrate preference was unchanged (*Figure 1h*). Animals moved less during nighttime measurements (*Figure 1i*). In absolute terms, B2D-fed mice consumed the same amount of food as controls (*Figure 1j*). Female mice exposed to B2D also showed reduced body weight gain and locomotory activity with trends toward lower oxygen consumption in the night phase (*Figure 1—figure supplement 2*). In contrast to B2D, reducing riboflavin in the diet by 90% did not impact liver FAD levels nor influence energy balance (*Figure 1—figure supplement*

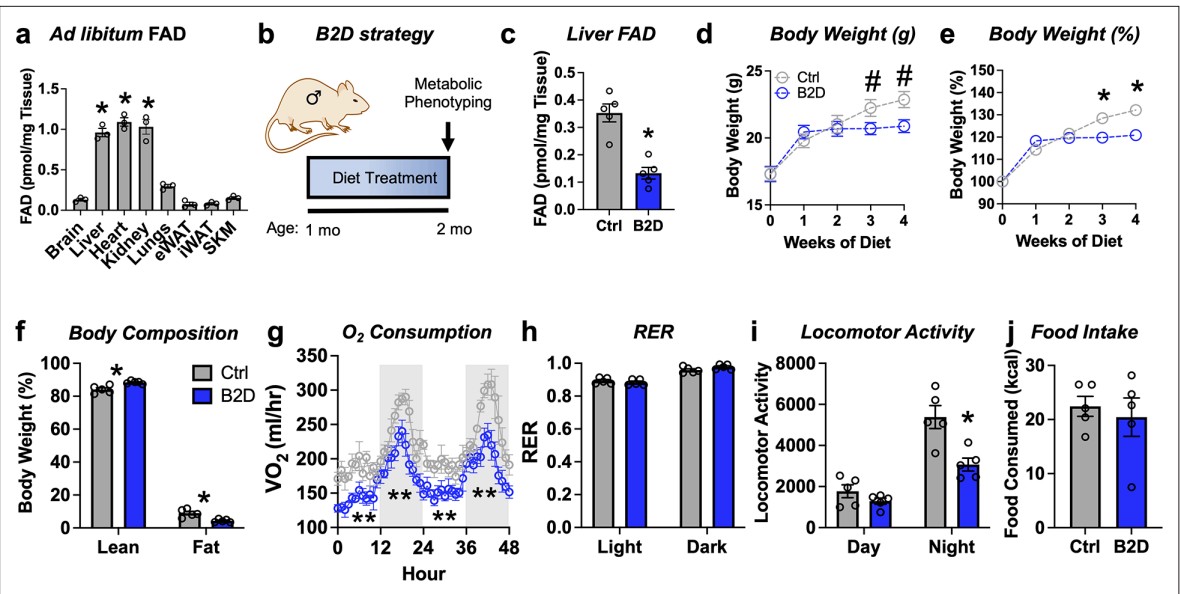

**Figure 1.** Riboflavin deficiency alters body composition and energy expenditure. (**a**) Ad libitum flavin adenine dinucleotide (FAD) concentrations measured from male WT mice (n = 3). (**b**) One-month-old male mice (n = 5 per group) were exposed to 99% vitamin B2-deficient diet (B2D) or isocaloric control diet (Ctrl) for 1 mo, followed by metabolic phenotyping. (**c**) FAD concentrations in the fasted liver. (**d**) Body weight (g), (**e**) % body weight gain and (**f**) body composition (% of body mass). Mice were individually housed and monitored in CLAMS-HC. (**g**) Recorded traces of oxygen consumption ($VO_2$), (**h**) respiratory exchange ratio (RER), (**i**) locomotor activity, and (**j**) cumulative food intake during dark (gray) and light (white) periods in the metabolic cages. One-way ANOVA with Sidak multiple-comparison tests was used for (**a**). Liver, heart, and kidney show statistically higher FAD levels versus other tissues. Other statistical analyses include two-way ANOVA with Sidak multiple-comparison tests for body weight (**d, e**) and body composition (fat and lean mass%) by Mann–Whitney for Ctrl v. B2D (**f**). Statistical analysis of CLAMS data was performed by ANCOVA with lean body mass as a co-variate (**g–j**). Data are represented as mean ± SEM. #$p<0.10$, *$p<0.05$, **$p<0.02$. Numerical data for individual panels are provided in *Figure 1—source data 1*.

The online version of this article includes the following source data and figure supplement(s) for figure 1:

**Source data 1.** Numerical data presented in *Figure 1*.

**Figure supplement 1.** Effects of 99% riboflavin deficiency on brown adipose tissue (BAT) in male and female mice.

**Figure supplement 1—source data 1.** Numerical data presented in *Figure 1—figure supplement 1*.

**Figure supplement 2.** Metabolic effects of 99% riboflavin deficiency in female mice.

**Figure supplement 2—source data 1.** Numerical data presented in *Figure 1—figure supplement 2*.

**Figure supplement 3.** Metabolic effects of 90% riboflavin deficiency in mice.

**Figure supplement 3—source data 1.** Numerical data presented in *Figure 1—figure supplement 3*.

*3*). These results identify B2 requirements that make FAD available for energy expenditure requirements and body weight in male and female mice.

## FAD is required for hepatic glucose production during fasting

Common clinical phenotypes of flavoprotein and nutritional riboflavin depletion disorders include fasting intolerance and hypoglycemia derived from impaired liver glucose production and metabolic flexibility (*Houten et al., 2016*). We found that liver FAD displayed circadian accumulation (*Patel et al., 2012*) coinciding with the onset of gluconeogenesis (ZT12) that occurs before the active period for mice (*Figure 2a*). Follow-up experiments also demonstrated re-feeding mice after 16 hr of fasting raised blood glucose levels (*Figure 2b*), but depleted liver FAD concentrations (*Figure 2c*). We next sought to understand whether the changes in FAD that occur in the mouse liver during B2D affected gluconeogenesis using multiple in vivo methods. After 4 wk of B2D, we subjected mice to a pyruvate tolerance test (PTT) after an overnight fast. Consistent with impaired gluconeogenesis from pyruvate, blood glucose concentrations were lower in B2D conditions compared with control diets at all times during the PTT (*Figure 2d*). Further plasma analysis established trends toward higher fasting concentrations of lactate, triglycerides (TG), free fatty acids (FFAs), and ketone bodies (*Figure 2—source*

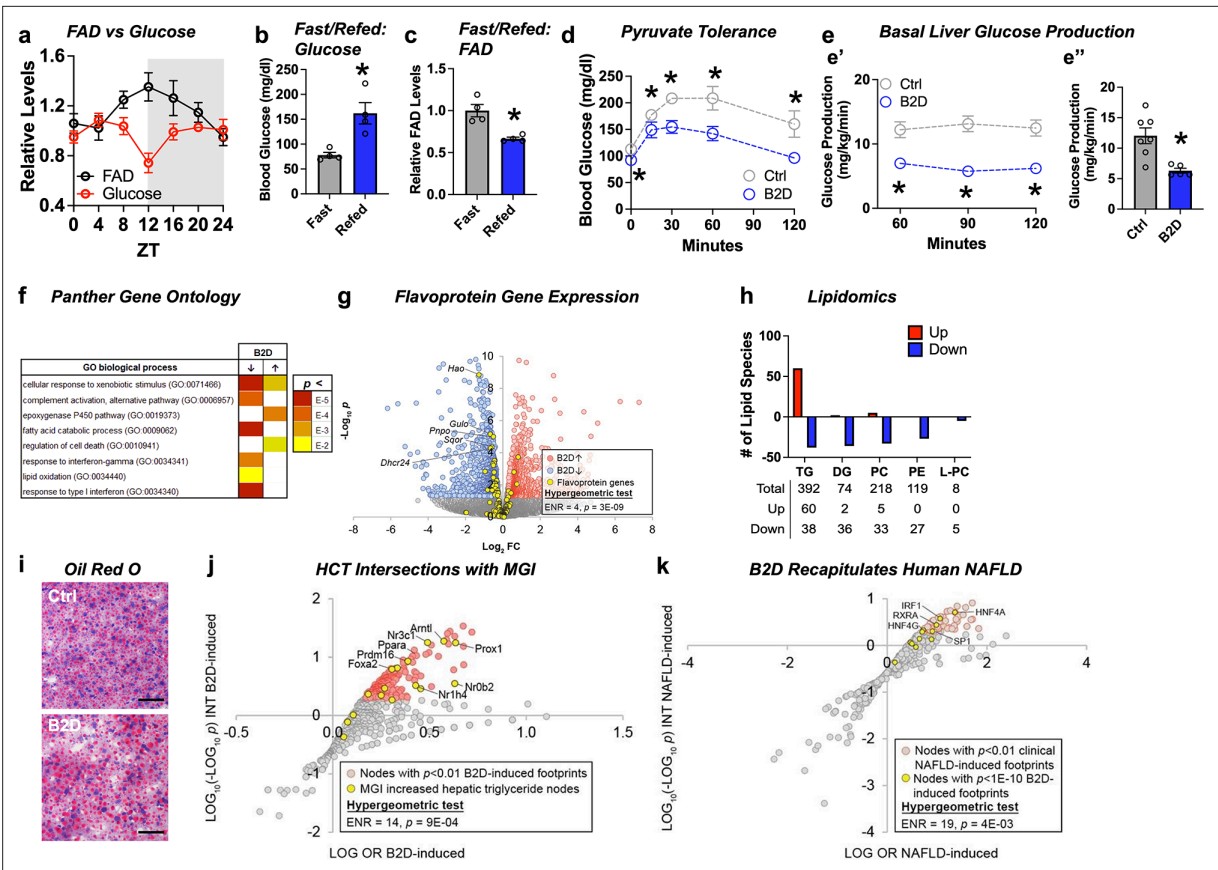

**Figure 2.** Liver glucose production requires bioavailable riboflavin. (**a**) Liver flavin adenine dinucleotide (FAD) and glucose levels across light/dark cycles (Zeitgeber Time, ZT) in male mice. Overnight (16 hr) fasting and re-fed (6 hr) (**b**) blood glucose and (**c**) liver FAD levels from male WT mice (n = 4). (**d**) Blood glucose excursion during pyruvate tolerance tests (n = 5 per group). (**e**) Basal glucose production was measured in fasted (18 hr) conscious mice (n = 5 vitamin B2-deficient diet [B2D] or n = 7 Ctrl) using constant rate of intravenous infusion (0.1 uCi/min) of [3-$^3$H]-glucose using a through a surgically implanted catheter in jugular vein. Whole-body basal glucose production rates (mg/kg/min) were calculated by dividing the [3-$^3$H] glucose infusion rate by the plasma glucose-specific activity corrected to the body weight. (**e′**) Steady-state rates were reached within 1 hr of infusion. (**e″**) Basal hepatic glucose production averaged across the three sampling times. (**f**) RNA-seq coupled with Panther Gene Ontology analysis identified pathways altered by B2D in the liver (n = 5 independent animals/diet). (**g**) Volcano plot depicting expression levels of flavoprotein genes (yellow) after B2D compared to control. (**h**) Lipidomic analysis of Ctrl and B2D (n = 5 per group). This analysis identified triglycerides significantly (p<0.05) increased in B2D versus Ctrl fed mice. (**i**) Representative Oil-Red-O stained liver sections from B2D or Ctrl. Scale bar 50 μm. (**j**) Enrichment of nodes encoded by genes that map to Mouse Genome Informatics (MGI) hypertriglyceridemia phenotypes among nodes with significant HCT intersections with B2D-induced genes. (**k**) Overlap of B2D-enriched nodes with nodes enriched in the human non-alcoholic fatty liver disease (NAFLD) gene expression consensome. The human NAFLD gene expression consensome ranks 18,162 genes based on their discovery across publicly archived clinical NAFLD case/control transcriptomic datasets. Statistical significance (*p<0.05) calculated by Mann–Whitney (**b, c, e″**) or two-way ANOVA with Sidak multiple-comparison test (**d, e′**). Statistical enrichment shown by hypergeometric test for (**j, k**). Data are represented as mean ± SEM. Numerical data for individual panels are provided in *Figure 2—source data 1*. Serum parameters in and liver lipid measurements are provided in *Figure 2—source data 2* and *Figure 2—source data 3*, respectively.

The online version of this article includes the following source data for figure 2:

**Source data 1.** Numerical data presented in *Figure 2*.

**Source data 2.** Serum parameters in vitamin B2-deficient diet (B2D) interventions.

**Source data 3.** Liver triglycerides and cholesterol in vitamin B2-deficient diet (B2D) interventions.

*data 2*). To further determine the effects of B2D on gluconeogenesis in vivo, we measured basal liver glucose production using an intravenous infusion of radiolabeled glucose in awake mice (*Chopra et al., 2008*; *Saha et al., 2004*). In a liver-specific way, B2D suppressed in vivo hepatic basal glucose production inferred from $^3$H-glucose infusions into fasted mice (*Figure 2e*). Steady-state basal glucose production rates were reached within 1 hr of infusion (*Figure 2e′*) and B2D reduced average rates

by more than 50% (*Figure 2e''*). These data indicate that FAD depletion directly affects liver glucose metabolism.

To explore molecular outcomes of riboflavin depletion, we used RNA-seq to identify biologically cohesive gene programs of B2D in the liver. Consistent with known roles of FAD in FAO, pathways related to fatty acid catabolism and lipid oxidation were strongly repressed in response to B2D (*Figure 2f*). Among the B2D-repressed genes annotated to the GO fatty acid catabolism pathway, we noted several encoded flavoproteins whose loss-of-function cause rare organic acidemias, including *Sqor* (*Friederich et al., 2020*) and *Ivd* (*Vockley and Ensenauer, 2006*). To further investigate the effect of B2D on flavoprotein gene expression, we curated a set of 117 genes encoding flavoproteins that require FAD or FMN for activity. Reflecting a specific impact of riboflavin deficiency, flavoprotein genes were enriched among B2D-repressed genes (*Figure 2g*) but not B2D-induced genes.

IEMs that arise from mutations in genes encoding mitochondria FAD transfer enzymes for FAO are identified by elevated organic acids in the blood, lipodystrophy, and fatty liver disease (*Balasubramaniam et al., 2019*). Lipidomics analysis in the liver identified phospholipid (phosphatidylethanolamine [PE], phosphatidylcholine [PC], and lyso-PC) and diacylglycerol (DG) as robustly attenuated (*Figure 2h*) in B2D-exposed mice compared to controls. Hepatic steatosis (*Figure 2i*) also accompanied B2D effects, presumably due to increased TG (*Figure 2—source data 3*). Accordingly, we hypothesized that altered expression of genes in the liver of B2D relative to normal diet controls reflected non-alcoholic fatty liver disease (NAFLD). To test this hypothesis, we performed high-confidence transcriptional target (HCT) intersection analysis (*Ochsner et al., 2019*) to identify signaling nodes with significant regulatory footprints among liver B2D-induced or -repressed genes. We validated the HCT footprint of B2D against mouse genes from the International Mouse Phenotyping Consortium, whose knockout increased liver TG (*Figure 2j*). From this approach, we identified strong enrichment of genes induced by B2D with metabolic transcription factor knockouts that cause fatty liver disease, such as *Ppara* (*Cotter et al., 2014*; *Kersten et al., 1999*; *Montagner et al., 2016*), *Nr1h4* (*Sinal et al., 2000*), and *Nr0b2* (*Huang et al., 2007*). Using other computational approaches to expand upon the underpinnings of macrosteatosis caused by B2D, we retrieved nodes previously shown to contain significant transcriptional footprints within genes differentially expressed in clinical NAFLD (*Bissig-Choisat et al., 2021*). B2D-induced genes consisted of footprints for transcription factor nodes active in NAFLD (*Figure 2k*), including HNF family members (*Xu et al., 2021*) and SREBP1 (*Shimano et al., 1997*). Collectively, our unbiased approach converged metabolic phenotypes and regulatory networks of B2D with those driving NAFLD and macrosteatosis observed in organic acidemias.

## PPARα activity maintains liver FAD pools for fasting glucose availability

Analysis of RNA-seq and ChIP-seq data (*Lee et al., 2014*; *Oshida et al., 2015*) discovered strongly enriched PPARα binding near promoter regions for 43 out of 121 putative flavoproteins (p=1.19 × $10^{-11}$, hypergeometric test) and supported direct coupling of FAD availability and nuclear receptor activity in the liver. Using DNA motif analysis, we also found canonical PPARα target genes among the genes downregulated by B2D (*Figure 3a*). Likewise, we identified a set of BioGRID-curated interaction partners of PPARα that was enriched among nodes with strong footprints in B2D-repressed genes (*Figure 3a*). Furthermore, B2D reduced PPARα-regulated flavoprotein genes (*Figure 3b*). The observation that patterns of B2D-sensitive gene expression in the liver overlap with PPARα regulatory footprints indicated a convergent role for PPARα and riboflavin in the transcriptional control of gluconeogenic responses.

To explore the physiological intersections between PPARα and B2D, we deleted *Ppara* in the liver by administering adeno-associated virus (AAV) expressing either Cre recombinase or a GFP under a liver-specific promoter to mice (AAV-TBG-Cre or AAV-TBG-GFP, respectively) harboring a floxed *Ppara* allele (*Ppara^flox/flox*). Cre-mediated excision of the loxP sites depletes *Ppara* in adult mice (*Ppara ΔHep*) and allows us to determine how PPARα mediates B2D responses in mice (*Figure 3c*). During 4 wk of diet intervention, *Ppara ΔHep* showed modest body weight gain that was slightly greater than B2D in magnitude. Combined B2D and knockdown of *Ppara* completely abolished body weight gain (*Figure 3d and e*) and reduced fat mass (*Figure 3f*) relative to control diet conditions. We also learned *Ppara ΔHep* largely recapitulated B2D effects on blood glucose excursion curves during PTTs (*Figure 3g and h*). Consistent with previous studies (*Stec et al., 2019*), *Ppara ΔHep* showed greater fat accumulation in the liver relative to *Ppara^flox/flox* on control diet. B2D strongly increased liver lipid

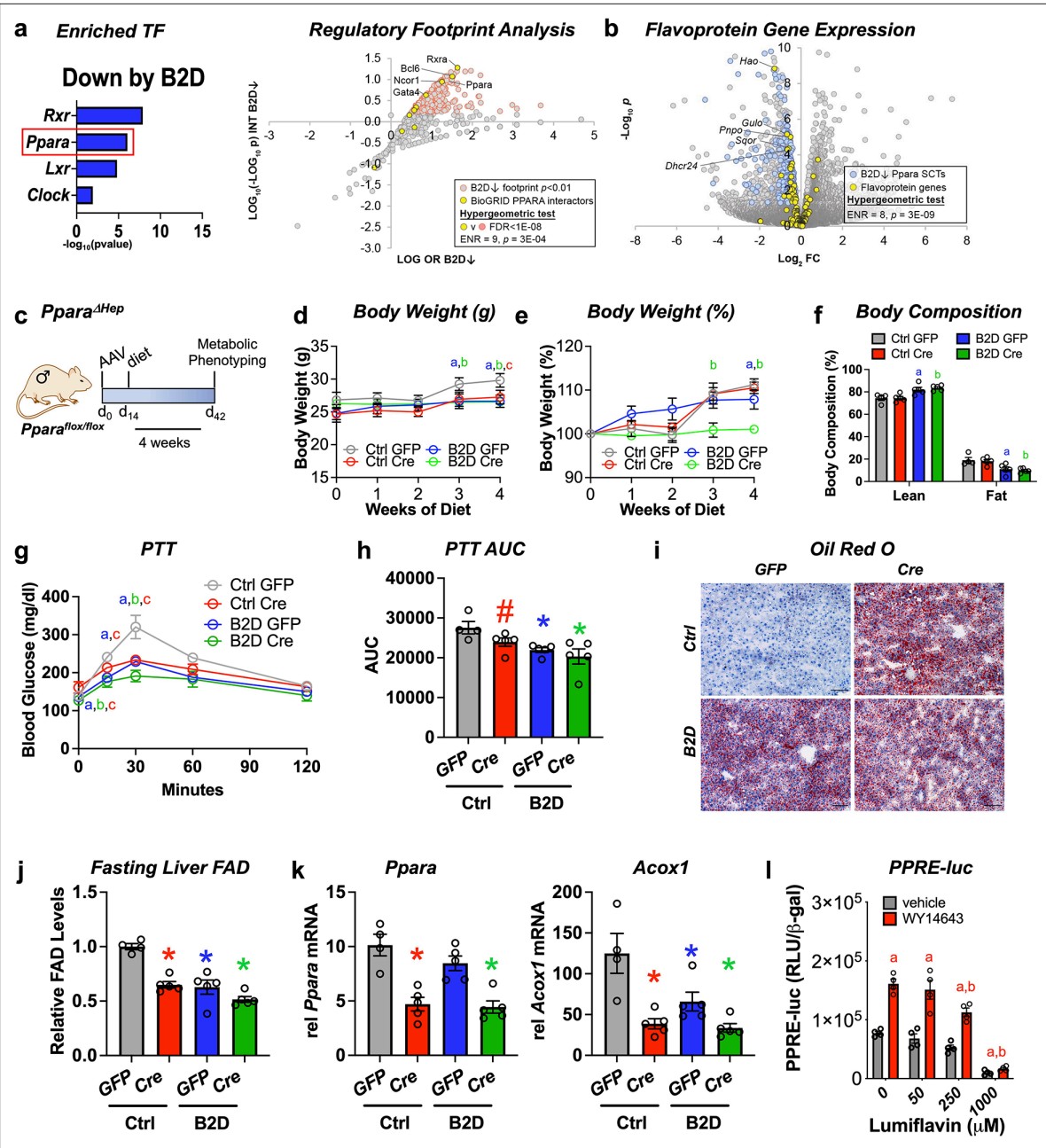

**Figure 3.** Liver PPARα governs glucogenic responses to dietary riboflavin. (**a**) Top enriched, vitamin B2-deficient diet (B2D) repressed gene sets using the EnrichR transcription factor collection (left). Scatterplot showing enrichment of known BioGRID-curated PPARα interacting nodes (yellow) among nodes with the most significant intersections with B2D-repressed genes (right). (**b**) Volcano plot depicting expression levels of PPARα-regulated flavoprotein genes (yellow) after Ctrl or B2D. (**c**) One-month-old *Ppara^flox/flox* male mice (n = 4–5 per group) received either AAV-TBG-GFP or AAV-TBG-Cre. Two weeks later, mice were provided 99% B2D or isocaloric diet (Ctrl) for 1 mo. (**d**) Body weight (g), (**e**) % body weight gain and (**f**) body composition (% of body mass). (**g**) Blood glucose levels and (**h**) area under the curve (AUC) during pyruvate tolerance tests. (**i**) Representative Oil-Red-O stained liver sections after B2D, scale bar 100 µm. (**j**) Flavin adenine dinucleotide (FAD) concentrations in the fasted liver after 4 wk of B2D. (**k**) Relative mRNA expression of *Ppara* and *Acox1*. *Tbp* served as the invariant control. (**l**) HepG2 cells were transiently transfected with a PPARα expression plasmid. PPRE luciferase reporter activity was measured in response to PPARα agonist (WY-14643) ± lumiflavin and normalized to β-galactosidase (n = 4). Statistics: *p<0.05 vs. control GFP (**a**, B2D GFP; **b**, B2D Cre; **c**, Ctrl Cre) by two-way ANOVA with Fisher LSD (**d, e, g**), *p<0.05 for body composition (fat and lean mass%) by Mann–Whitney vs. control GFP, *p<0.05, #p<0.10 vs. control GFP by one-way ANOVA with Dunnet's test and Fisher LSD (**h, j, k**). *p<0.05 by two-way ANOVA with Tukey's multiple-comparison test for (**l**), a and b are statistical differences vs. vehicle and WY-14643, respectively. Statistical enrichment shown by hypergeometric test (**a**, right, **b**). Data are mean ± SEM. Numerical data for individual panels are provided in *Figure 3— source data 1*.

*Figure 3 continued on next page*

*Figure 3 continued*

The online version of this article includes the following source data and figure supplement(s) for figure 3:

**Source data 1.** Numerical data presented in *Figure 3*.

**Figure supplement 1.** *Ppara* whole-body knockout responses to dietary riboflavin depletion.

**Figure supplement 1—source data 1.** Numerical data presented in *Figure 3—figure supplement 1*.

across genotypes (*Figure 3i*), demonstrating parallel effects of B2D and *Ppara ΔHep* on liver energy balance. Additionally, FAD levels were lowered to similar extents by *Ppara ΔHep* or B2D relative to *Ppara^flox/flox* on control diet (*Figure 3j*). We also phenotyped male *Ppara* whole-body knockout (pKO) mice exposed to control or B2D for 1 mo (*Figure 3—figure supplement 1*). As expected (*Cotter et al., 2014*), we learned pKO largely negated effects of B2D on body weight gain. PTT demonstrated B2D and pKO decreased the conversion of glucose release relative to control diets and wild-type controls.

To extend our studies of gene regulation, we profiled *Ppara* and the PPARα target gene *Acox1*. *Ppara* levels were reduced in *Ppara ΔHep*, along with *Acox1* (*Figure 3k*). Furthermore, the blunted expression levels of *Acox1* to diet and genetic interventions (*Ppara ΔHep*) confirm B2D confers diminished PPARα activity in the mouse liver. In a complementary approach, we performed experiments to determine how FAD depletion affects PPARα transcriptional activity using the riboflavin analog and competitive inhibitor lumiflavin (*Figure 3l*). We found lumiflavin reduced PPRE-luciferase activity in the presence of the PPARA agonist WY-14643, suggesting intact FAD pools contribute to PPARα transcriptional activity. Altogether, these findings suggest that PPARα sustains FAD levels and requires riboflavin to direct glucogenic responses in the liver.

## PPARα activation by fenofibrate rescues liver glucose production even when FAD cannot be generated from diet

The FDA-approved fibrate drugs act selectively on PPARα to lower blood lipids and treat hypertriglyceridemia (*Bougarne et al., 2018*). PPARα agonists also show promise for the treatment of some mitochondrial disorders (*Steele et al., 2020*). The convergence of B2D effects on PPARα regulation of gene expression and metabolic responses suggested fenofibrate treatment may restore some metabolic competency in animals on riboflavin-deficient diet. To explore this possibility, we administered fenofibrate after 7 wk of B2D exposure (*Figure 4a*). Daily gavage with fenofibrate (300 mg/kg) for 2 wk while maintaining mice on diet interventions significantly reduced body weight gain under B2D conditions (*Figure 4b and c*). At the end of the experiment, fasted mice received fenofibrate 2 hr before blood glucose measurements. Fenofibrate increased blood glucose in both groups of mice far above pre-gavage levels (*Figure 4d*). When liver histology was examined, we did not detect hepatic steatosis in B2D mice that received fenofibrate (*Figure 4e*).

Given the ability of PPARα to regulate flavoprotein gene expression, we pursued additional RNA-Seq studies to understand the mechanisms that allowed fenofibrate to rescue hypoglycemia in B2D. Consistent with restoration of the flavoprotein transcriptome in response to PPARα activation, we found the number of flavoprotein genes induced by B2D+ fenofibrate more than doubled when compared to B2D alone (*Figure 4f*). Given the improvement in hepatic steatosis after PPARα activation, we hypothesized the B2D+ fenofibrate treatment caused inversion of the alignments between B2D and NAFLD transcription networks. Using this approach, we found gene footprints enriched by B2D+ fenofibrate and those depleted in clinical NAFLD converged (*Figure 4g*), including the GABP transcriptional program inactivated in inflammatory liver diseases (*Niopek et al., 2017*). These unbiased approaches strengthen the notion that PPARα activation overcomes fatty liver and hypoglycemia phenotypes imposed by B2D.

## Altered sphingolipid pools and respiratory chain efficiency in B2D

Fat and protein metabolism produces substrates for the synthesis of sphingolipids, such as ceramides and dihydroceramides, whose tissue accrual associates with severity of fatty liver disease (*Luukkonen et al., 2016*) and mitochondrial dysfunction (*Hammerschmidt et al., 2019*; *Park et al., 2016*). To determine whether B2D or B2D+ fenofibrate also changed the sphingolipid composition of the mouse liver, we measured a battery of sphingolipids and detected varying levels of multiple

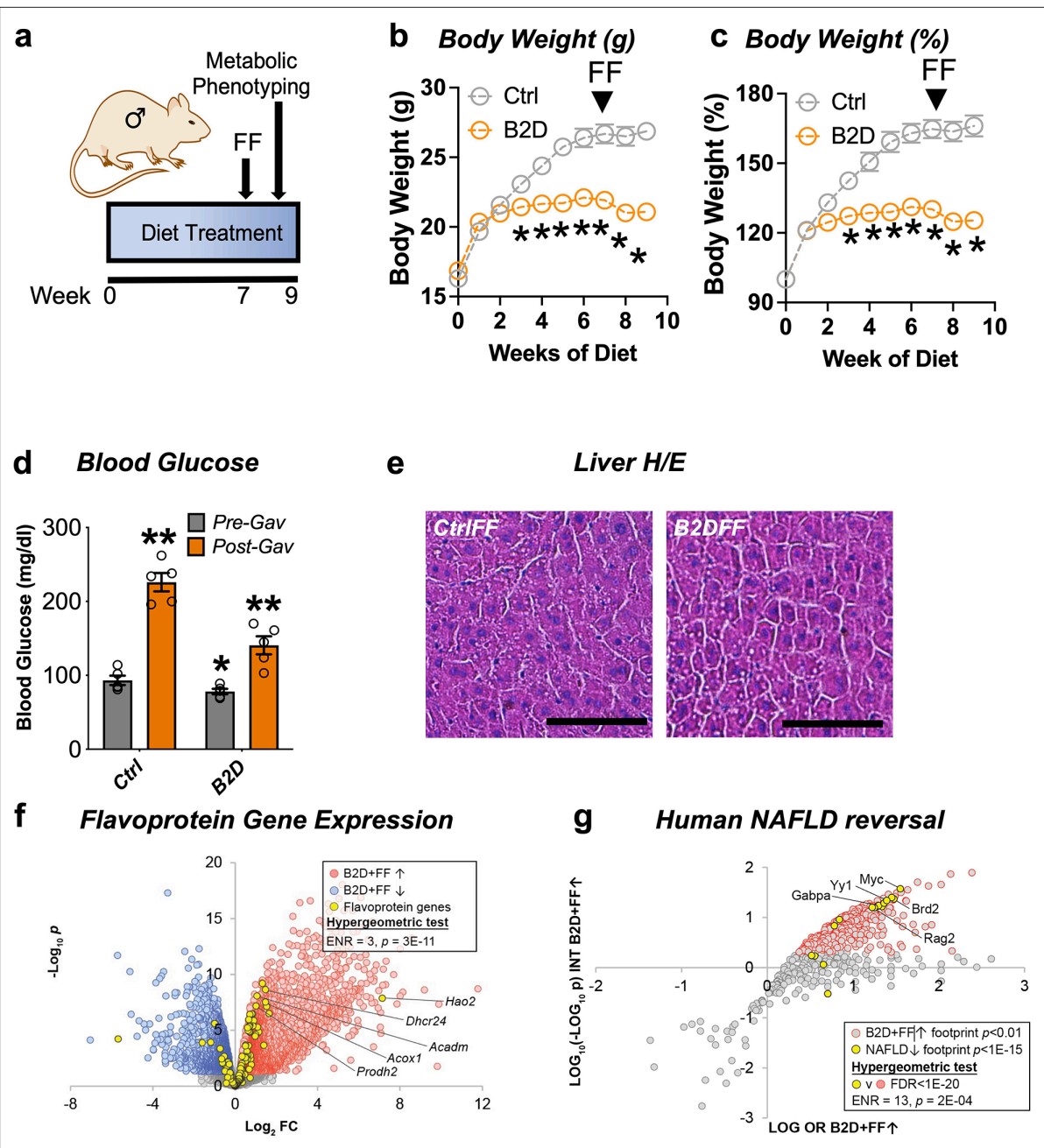

**Figure 4.** Remnant PPARα activation rescues fasting responses impaired by vitamin B2-deficient diet (B2D). (**a**) One-month-old male mice (n = 10 per group) were exposed to 99% vitamin B2-deficient diet (B2D) or isocaloric control diet (Ctrl) for 9 wk. For weeks 7–9, mice were gavaged daily with fenofibrate (FF). (**b**) Body weight (g) and (**c**) body weight gain during treatments. For post-gavage (week 9), mice were fasted overnight and administered FF 2 hr before measurements. (**d**) Overnight fasting blood glucose levels for pre-gavage (gray) and after 2 wk of FF treatment (orange). (**e**) Representative H&E-stained liver sections from Ctrl or B2D mice following FF treatment. Scale, 100 μm. (**f**) Expression level of flavoprotein genes in B2D-fed mice following FF. (**g**) Enrichment of B2D+FF with gene footprints repressed in human NAFLD. Statistical analyses include two-way ANOVA with Sidak multiple-comparison test (**b–d**) and *p<0.05 for (**b, c**). *p<0.05 pre-gavage Ctrl vs. B2D or **p<0.05 post-gavage vs. pre-gavage were p-value cutoffs for (**d**). Statistical enrichment shown by hypergeometric test (**f, g**). Data are represented as mean ± SEM. Numerical data for individual panels are provided in *Figure 4—source data 1*.

The online version of this article includes the following source data for figure 4:

**Source data 1.** Numerical data presented in *Figure 4*.

species altered by diet and fenofibrate treatments (*Figure 5—figure supplement 1*). Pool sizes for deoxysphingolipids that require alanine condensation with palmitoyl-CoA trended higher with B2D (*Figure 5a*). This finding may derive from the serine availability during hypoxic stress and higher NADH levels (*Yang et al., 2020*). Consistent with this idea, B2D conditions reduced the NAD/NADH ratio in the liver (*Figure 5b*). Notably, deoxysphingolipids impair mitochondrial function (*Alecu et al., 2017*; *Muthusamy et al., 2020*) and lead to an energy deficit, especially in the liver, where fasting requires elevated demand for biomass.

Pathogenic variants in genes for riboflavin transport and metabolism that deplete FAD impair electron transfer from flavoenzymes in the ETC to coenzyme $Q_{10}$ and, ultimately, the energy requirements for gluconeogenesis and fasting tolerance (*Rinaldo et al., 2002*). During prolonged fasting, beta-oxidation and proteolysis produce high fluxes of electrons that flow through the ETC. Coenzyme $Q_{10}$ collects and converges electron flow on complex III via complex II (oxidizing succinate into fumarate) or complex I. In this way, the coenzyme $Q_{10}$ pool accommodates variable electron fluxes and manages the mitochondria redox environment. Analysis of FAD and FMN (*Figure 5c*) confirmed fenofibrate was incapable of reconciling cofactor pools and lacked meaningful impacts on relative levels of some oxidative phosphorylation proteins (*Figure 5d*). In contrast to the reduced NAD/NADH, we found B2D treatments accumulated oxidized coenzyme $Q_{10}$ and rodent-biased coenzyme $Q_9$ (*Figure 5e*), suggesting higher $Q/QH_2$, more negative free energy for complexes I/II, and compensation for the defects in mitochondrial energy efficiency associated with B2D (*Satapati et al., 2015*).

## Fenofibrate activates the integrated stress response in B2D

We next measured concentrations of carnitines, amino acid, and hydrophilic metabolites in the liver using mass spectroscopy across the B2D experiments. B2D altered the steady-state levels of valine, methionine, phenylalanine, and caused accumulation of short-chain C5 carnitines that reflect incomplete beta-oxidation of fatty acids (*Figure 6a*). Upper arms of glucose metabolism (3PG/2PG) showed lower activities coupled with higher lactate and buildup of metabolites above pyruvate oxidation (G3P, PEP). Metaboanalyst (*Pang et al., 2021*) revealed complete removal of dietary riboflavin caused metabolite changes in the liver that enriched for organic acidemias and inborn errors of the TCA cycle (*Figure 6b*). Moreover, B2D caused accumulation of oxidized TCA cycle metabolic intermediates, including fumarate, malate, and 2HG. These findings were compatible with gene sets (*Figure 6c*) associated with hypoxia and epithelial-mesenchymal transition (*Sciacovelli et al., 2016*; *Ward et al., 2010*).

PPARα activity favors conversion of proteins to provide amino acids as substrates for anabolic processes (*Kersten et al., 2001*). Our phenotyping analysis of fenofibrate suggested alternative carbon sources might supply the substrates to support glucose production during B2D (*Figure 4c*), including pyruvate, which recovered blood glucose in B2D to pre-gavage control levels (*Figure 6d*). Amino acids, such as alanine and serine, are also significant contributors to de novo synthesis of glucose. In line with this idea, gluconeogenic amino acids (serine, glutamate, histidine, alanine) showed selective and unique accumulation in the combined B2D and fenofibrate treatments (*Figure 6a*). Steady-state levels of other anaplerotic amino acids that replenish the TCA cycle (*Figure 6a*), isoleucine, and threonine, also increased at the expense of reduced ketosis and β-hydroxybutyrate (*Figure 2—source data 3*). Similarly, B2D+ fenofibrate elevated levels of TCA cycle metabolites citrate and aconitate and increased the fumarate to succinate ratio. Importantly, we noted moderated levels of carnitines enriched in organic acidemias (C5 and C6). These conditions suggest PPARα activation inhibits cataplerosis of TCA cycle intermediates.

The constellation of phenotypic effects resulting from riboflavin depletion and fenofibrate suggested global shifts in gene expression and metabolism. RNA-seq indicated B2D+ fenofibrate precipitated a response with elements of greater amino acid and glucose metabolism (*Figure 6e*). We also observed a gene signature for the integrated stress response (ISR) further supported by increased expression of the master transcription factor regulator *Atf4,* as well as its key target genes (*Psat1*, *Gls*, *Fgf21*, *Gpt2*, *Ddit4*) during combined B2D+ fenofibrate treatments. Mechanistically, ISR activation occurs through phosphorylation of eIF2α and ATF4 translation, which mediates transcription of target genes to resolve the ISR and regain amino acid homeostasis (*Harding et al., 2003*; *Ye et al., 2012*). In line with this idea, B2D+ fenofibrate achieved robust GCN2 expression, eIF2α phosphorylation, and higher ATF4 levels relative to any other treatment (*Figure 6f*). Together, our integrated transcription

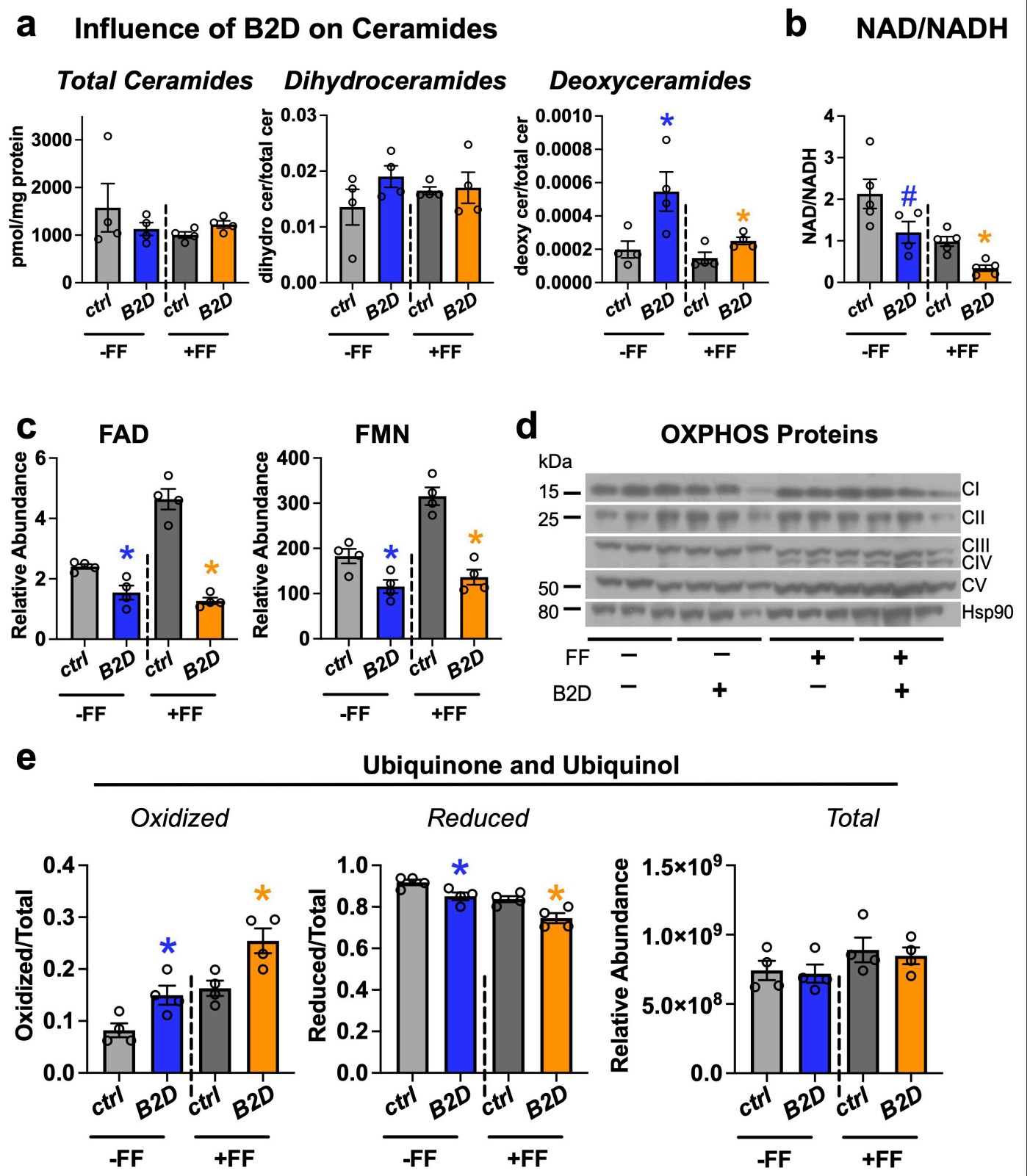

**Figure 5.** Fenofibrate impacts mitochondrial respiratory chain efficiency but does not rescue flavin adenine dinucleotide (FAD) levels in vitamin B2-deficient diet (B2D). (**a**) Fasting liver sphingolipid species (pmol/mg protein) and pool sizes by mass spectrometry (n = 4 per group). (**b**) Hepatic NAD/NADH ratio. (**c**) Fasting liver FAD and flavin mononucleotide (FMN) relative abundance by mass spectrometry (n = 4 per group). (**d**) Western blot analysis of oxidative phosphorylation complexes. HSP90 served as the invariant control. (**e**) Ratio of oxidized and reduced liver coenzyme Q10 and Q9 by mass

*Figure 5 continued on next page*

*Figure 5 continued*

spectrometry (n = 4 per group). Statistics are Mann–Whitney test for Ctrl vs. B2D (-FF) or Ctrl vs. B2D (+FF) for (**a, b, c, e**). *p<0.05, #p<0.10. Data are represented as mean ± SEM. Numerical data for individual panels are provided in *Figure 5—source data 1*. Full gel images are provided in *Figure 5— source data 2*.

The online version of this article includes the following source data and figure supplement(s) for figure 5:

**Source data 1.** Numerical data presented in *Figure 5*.

**Source data 2.** Full gel images and original image files for western blots in *Figure 5*.

**Figure supplement 1.** Vitamin B2-deficient diet (B2D) effects on sphingolipids in the liver.

**Figure supplement 1—source data 1.** Numerical data presented in *Figure 5—figure supplement 1*.

and mass spectrometry analyses favor a model in which PPARα activation precipitates concerted activation of the ISR, which in turn overcomes the fasting intolerance imposed by B2D (*Figure 6g*).

## Discussion

Results presented here reveal that vitamin B2 provides critical substrates for glucose availability during fasting in mice. Mammals cannot synthesize vitamin B2, so diet remains the only available source of FAD and FMN (*Powers, 2003*). Our results show that 4 wk of 90% B2D did not influence whole-body metabolic phenotypes. Based on data reported by *Fenton and Cowgill, 1947*, mice require about 4 mg riboflavin/kg diet for normal growth. However, mouse chow generally contains almost twice the daily riboflavin requirement (*Anonymous, 1977*). Most tissues store very little of the vitamin as riboflavin. The majority of bioavailable riboflavin is converted to FAD and excess is excreted in urine as FMN (*Rivlin, 1970*). Riboflavin turnover rates in rodents are also relatively slow at approximately 16 d (*Yang and McCormick, 1967*), which underscores sufficient FAD concentrations remain in the liver after 4 wk of 90% B2D. Therefore, the historical literature suggests 10% of the required level of riboflavin may be sufficient for growth and metabolism in the mouse during our 4-week monitoring period.

Our work sheds light on how the liver copes with severe metabolic crises and the FAO disorders caused by flavoprotein disruption or FAD depletion during prolonged fasting when both beta-oxidation and gluconeogenesis are concomitantly activated. We speculate these FAO disorders rely on conservation responses when the mitochondria are starved of FAD and FMN required for ETC activity. B2D reduces fat stores and slows metabolism for the energy conservation functions that adapt the animal to survive FAD and FMN depletion. Despite lower body weight gain when compared to control diet, B2D causes reduced energy expenditure. IEMs that arise from mutations in genes encoding mitochondrial FAD transfer enzymes for FAO are often identified by ectopic fat distribution and liver disease (*Balasubramaniam et al., 2019*). People with long-chain fatty acid oxidation disorders (LC-FAODs), including mutations in the FAD-dependent protein ACADVL, also show similar energy expenditure phenotypes as B2D. Indeed, energy expenditure is significantly lower among subjects with LC-FAODs compared to the reference population in the National Health and Nutrition Examination Survey (*DeLany et al., 2023*). These observations suggest that responses to B2D are relevant to studying IEMs defined by mutations in FAD-dependent metabolic enzymes.

Interestingly, the metabolic phenotype of B2D resists conditions unfavorable for mitochondrial function, including hypoxia, suggesting that these changes select for survival. Increased reliance upon glycolysis occurs in mitochondrial disease (*Robinson, 2006*) and hypoxia responses activate glycolytic enzymes that allow energy production when the mitochondria are starved of oxygen as a substrate for oxidative phosphorylation. Likewise, our data support the idea that the stress of B2D gives rise to an environment for fenofibrate and PPARα activation to co-opt the ISR, adapt to TCA cycle dysfunction, and engage hypoxia enzymes to reconcile anaplerosis. Elevated TCA cycle intermediates, such as fumarate, then act to stabilize antioxidant transcription factors and protect against oxidative stress in the liver (*Ashrafian et al., 2012*) during the stress of IEMs modeled in our study.

IEMs modeled in our study frequently present NAFLD-like phenotypes that contribute to fasting intolerance (*Rinaldo et al., 2002*). In our model, B2D alters lipid profiles and gene expression patterns that converge B2D with more common NAFLD phenotypes. The liver dominates mass-specific metabolic rates (*Rolfe and Brown, 1997*) and, for this reason, the NAFLD caused by B2D likely reflects a

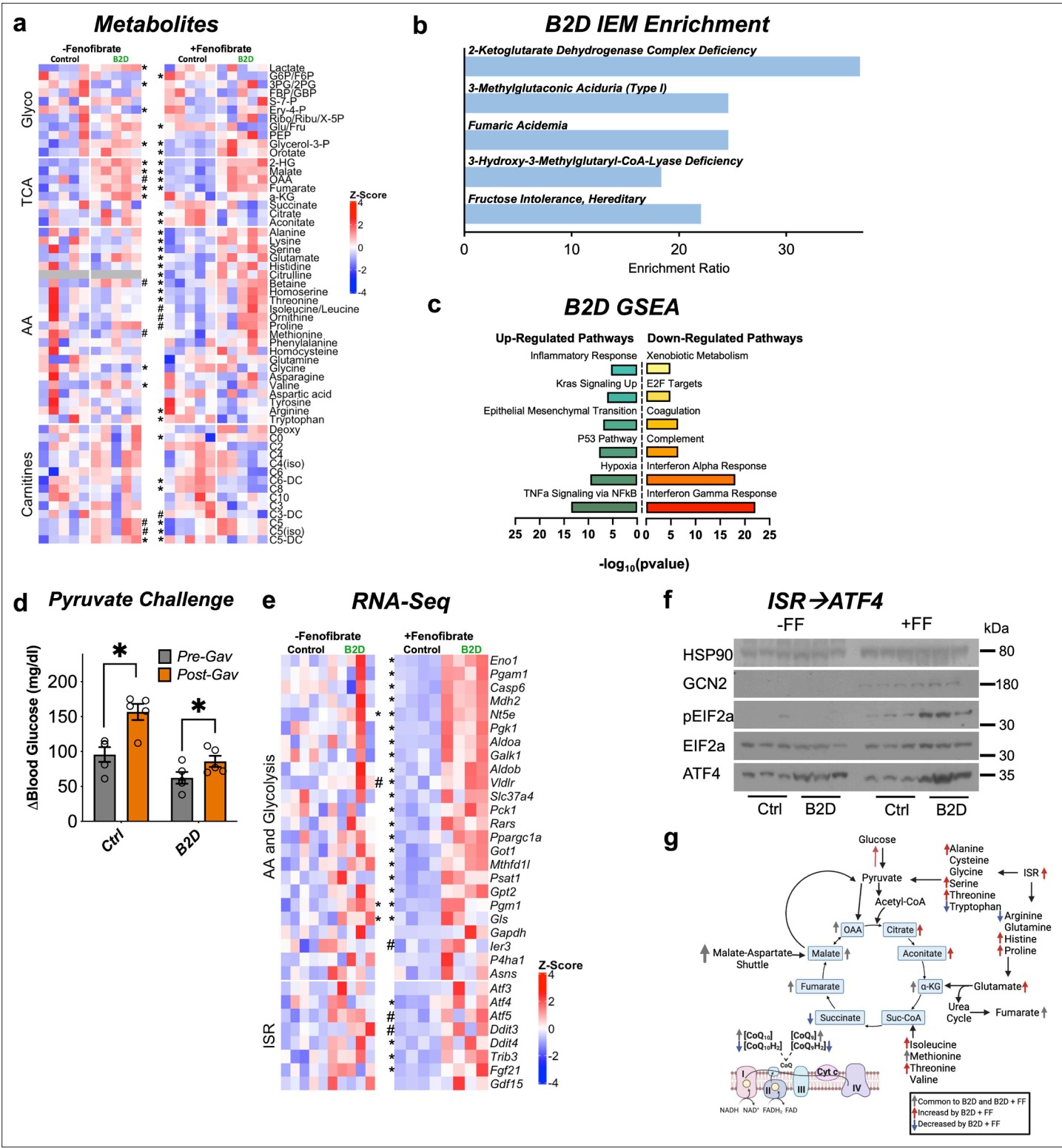

**Figure 6.** Integrated stress response (ISR) activity forms the basis to reconcile flavin adenine dinucleotide (FAD) disruption. (**a**) Glycolysis, tricarboxylic acid (TCA), amino acid, fatty acid oxidation (carnitines) metabolites measured by mass spectrometry across vitamin B2-deficient diet (B2D) and FF treatments (shown as Z-score). (**b**) Metaboanalyst integration demonstrates B2D causes organic acidemias. (**c**) RNA-seq coupled with Gene Set Enrichment Analysis (GSEA) identified gene signatures altered by B2D in the liver (n = 5 independent animals/diet). (**d**) Change in glucose levels 30 min after pyruvate injection pre- and post-fenofibrate gavage (n = 5). (**e**) Amino acid, glucose metabolism, and ISR genes in B2D or B2D + FF, shown as Z-score from RNA-seq data. (**f**) Western blot analysis for ISR proteins in liver lysates from mice treated with B2D (4 wk total) or B2D+FF (9 wk total)

*Figure 6 continued on next page*

*Figure 6 continued*

and fasted overnight. (**g**) B2D + FF activates the ISR, increasing amino acids that restore glucose availability. Statistical significance was assessed by Mann–Whitney (**a, e**) and two-way ANOVA and Fisher LSD (**d**). Data are represented as mean ± SEM. *p<0.05, #p<0.10. Numerical data for individual panels are provided in *Figure 6—source data 1*. Full gel images are provided in *Figure 6—source data 2*.

The online version of this article includes the following source data for figure 6:

**Source data 1.** Numerical data presented in *Figure 6*.

**Source data 2.** Full gel images and original image files for western blots in *Figure 6*.

combination of lower hepatic fatty acid oxidation contributing to the accumulation of liver triglycerides and other complex lipid species observed in obesity (*Shi et al., 2022*). Our lipidomics also revealed that B2D caused accumulation of deoxysphingolipids, which also become more abundant in NAFLD from incomplete fat oxidation and accrual of toxic intermediates (*Gai et al., 2020*). These findings are particularly relevant for discovering new biomarkers for fatty liver disease.

The energetic requirements of fasting dictate substrate oxidation and electron transport. B2D restricts mitochondrial function, which causes reductive pressures that likely exacerbate ROS formation in part through proton leak. PPARα activation prevents oxidative stress (*Ip et al., 2004*) by merging lower NAD/NADH and oxidized Q, which, in turn, lowers the free energy of electron transport through complexes I and II. These data are consistent with inhibition of complex II, resulting in a more oxidized Q pool and reveals an important adaptation that allows the liver to overcome riboflavin deficiency (*Treberg et al., 2011*).

Inefficient amino acid availability, or increased requirements of amino acids to maintain gluconeogenesis, activate ATF4 and the ISR. ATF4 is the principal downstream effector of the ISR, whose regulation becomes altered in human and rodent NAFLD (*Puri et al., 2008*; *Seo et al., 2009*). Integrated metabolomic and RNA-seq studies demonstrated that fenofibrate unveils the ISR to increase the abundance of anaplerotic amino acids. We are unaware of other studies that observe coincident activation of PPARα and ATF4 to drive adaptive responses to liver stress. However, induction of shared PPARα and ATF4 targets, including *Fgf21*, may contribute to hepatoprotection from lipotoxic lipid accumulation (*Montagner et al., 2016*). It will be interesting in the future to determine why B2D exposes a vulnerability to fenofibrate that engages selective genome regulation by PPARα and ATF4 for recovering glucose production in settings of fasting intolerance.

This study is unique in its comprehensive approach to understand the complex metabolic consequences of FAD depletion and riboflavin auxotrophy in mouse models. Nevertheless, we acknowledge some limitations of the study. While the effect of B2D may act through different pathways and tissues, we demonstrated some of these effects may be mediated through disturbance of nuclear receptor activity and altered glucose availability in the liver. A limitation of our study is that we have focused on how PPARα requires vitamin B2 in the mouse liver to produce glucose, and it is possible that other fasting responsive transcription factors (*Bideyan et al., 2021*) are sensitive to B2D. One important caveat of these experiments is that the kidney also contributes to glucose production during fasting (*Joseph et al., 2000*). Although our in vivo experiments do not explore whether the kidney reconciled any liver gluconeogenic deficiency, our observations reveal fundamental vitamin requirements to source the liver with the FAD and FMN pools necessary for energy balance.

There is evidence fibrates generate beneficial effects in IEMs, including improvement of antioxidant defenses and resolution of inflammation (*Seminotti et al., 2023*). While further studies are needed, we describe allostatic outcomes of PPARα activation that overcome bioenergetic costs of fasting. Rare diseases of flavoprotein mutations, including organic acidemias, cause substantial morbidity and have no cure. Therefore, understanding how nuclear receptor regulation of flavoprotein function and FAD pool distribution surmounts hypoglycemia and fatty liver is valuable for describing fundamental physiology and implementing future therapeutic strategies.

## Materials and methods
### Mice and housing conditions

All animal procedures were approved by the Institutional Animal Care and Use Committee of Baylor College of Medicine (AN-6411). All mice were housed in a barrier-specific pathogen-free animal facility

with 12 hr dark-light cycle and free access to water and food. C57BL/6J wild-type mice (RRID:IMSR_JAX:000664) were obtained from the BCM Center for Comparative Medicine and global *Ppara*[-/-] mice (RRID:IMSR_JAX:008154) were generated previously (*Lee et al., 1995*). In all experiments, male mice were randomly placed on control or riboflavin deficient diet starting at 4 wk of age. Riboflavin-deficient and matched control diets (*Xu et al., 2018*) were provided by Research Diets: 90% control, D10012G; 90% riboflavin-deficient, D12030102; 99% control, A18041301; or 99% riboflavin-deficient, A19080901. All diets were isocaloric, and amino acids were kept constant (*Supplementary files 1 and 2*).

## AAV TBG administration

For AAV-TBG injections, 1-month old *Ppara*[flox/flox] mice (*Brocker et al., 2017*) were injected with AAV8-TBG-iCre-WPRE (1 × 10[11] vc/ml, Addgene# 192878) or AAV8-TBG-EGFP-WPRE (1 × 10[11] vc/ml, Addgene #192880) into the tail vein in a total volume of 100 μl saline. These viruses were provided by the Gene Vector Core at Baylor College of Medicine. Mice were allowed to recover for 2 wk to permit for viral expression and recombination of floxed *Ppara* alleles before being used for B2D experiments.

## Fenofibrate gavage

We used malnutrition (*van Zutphen et al., 2016*), *Ppara* knockout (*Montagner et al., 2016*), and cancer cachexia (*Goncalves et al., 2018*) studies to design fenofibrate interventions. 0.5% methylcellulose solution was prepared by heating 150 ml water with 2.5 g 400 cP methylcellulose (Sigma, #M0262) added with stirring. Chilled water was added (350 ml) and stirred overnight at 4°C. Fenofibrate-gavage solution was made with 112.5 mg fenofibrate (Sigma, #F6020) in 1.5 ml 0.5% methylcellulose solution. Mice were gavaged daily at 300 mg/kg.

## Pyruvate tolerance test (PTT)

To determine pyruvate tolerance, mice were fasted for 16 hr, and Na-pyruvate was administered (1 g/kg body weight) by intraperitoneal injection. Blood glucose levels were monitored at 0, 15, 30, 60, and 120 min by a Freestyle Glucose Monitoring System (Abbott Laboratories).

## Basal glucose production rate in live mice

Basal glucose production was measured as described previously (*Saha et al., 2004*). In brief, mice were anesthetized, and a midline neck incision was made to expose the jugular vein. A microcannula was inserted into the jugular vein, threaded into the right atrium, and anchored at the venotomy site. Mice were allowed to recover for 4–5 d with ad libitum access to water and food. Following an overnight fast, the conscious mice received a bolus dose (10 uCi) and then a constant rate intravenous infusion (0.1 uCi/min) of chromatography-purified [3-$^3$H]-glucose using a syringe infusion pump for 120 min through surgically implanted catheter in jugular vein. For determination of basal glucose production, blood samples were collected after 0, 60, 90, and 120 min of labeled glucose infusion to calculate the basal glucose production rates using plasma glucose-specific activity of tritiated labeled glucose (3-$^3$H). Steady states were reached within 1 hr of infusion. Plasma was separated and deproteinized using equal volumes of barium hydroxide and zinc sulfate, dried to remove $^3$H$_2$O, resuspended in water, and counted in scintillation fluid using a liquid scintillation counter (Beckman Instruments, Palo Alto, CA). We calculated basal whole-body glucose production (mg/kg/min) by dividing the [3-$^3$H] glucose infusion rate by the plasma glucose-specific activity corrected to body weight (*Wall et al., 1957*).

## Indirect calorimetry

Wild-type mice were maintained on experimental diets and housed at room temperature in Comprehensive Lab Animal Monitoring System Home Cages (CLAMS-HC, Columbus Instruments). Oxygen consumption, $CO_2$ emission, energy expenditure, food and water intake, and activity were measured for 5 d (BCM Mouse Metabolic and Phenotyping Core). Mouse body composition was examined by magnetic resonance imaging (Echo Medical Systems) before indirect calorimetry.

## Histology

Sections of liver tissue were frozen in Tissue-Tek OCT compound (4583; Sakura Finetek USA), and neutral lipids stained with Oil Red O. Formalin-fixed paraffin-embedded tissue sections were stained with hematoxylin and eosin (H/E). Images were captured on a Nikon Ci-L Brightfield microscope.

## Serum and lipid assays

Fasted serum was used to measure serum lactate (K607; Biovision), ALT (TR71121; Thermo Scientific), AST (TR70121; Thermo Scientific), beta-hydroxybutyrate (Biovision K632), and serum-free fatty acids (#sfa-1; Zen-Bio).

## Liver FAD measurements

10–15 mg of liver tissue was deproteinated (K808; Biovision), followed by measurement of FAD through a colorimetric assay (K357; Biovision). FAD concentration was standardized to input tissue weight.

## Hepatic TG and cholesterol

Both serum and tissue samples were analyzed for triglycerides (Triglyceride reagent TR22421; Thermo Scientific) and cholesterol (Total Cholesterol Reagent TR13421; Thermo Scientific). Hepatic TGs and cholesterol were assayed as described previously (*Kim et al., 2019*). Briefly, liver homogenates were mixed with a 1:2 chloroform:methanol solution followed by isolation of the lipid-rich chloroform layer (modified Folch method).

## Immunoblotting

Cell and tissue lysates were prepared in Protein Extraction Reagent (Thermo Fisher) supplemented with Halt Protease and Phosphatase Inhibitor Cocktail (Thermo Fisher). Western blotting was performed with whole-cell lysates run on 4–12% Bis-Tris NuPage gels (Life Technologies) and transferred onto Immobilon-P Transfer Membranes (Millipore), followed by antibody incubation. Immunoreactive bands were visualized by chemiluminescence. Antibodies used in this study are listed in *Supplementary file 3*.

## RNA extraction and RNA-seq analysis

Total liver RNA was extracted using the QIAGEN RNeasy Plus Mini kit (74034; QIAGEN). Sample quality was confirmed on an Agilent 2100 Bioanalyzer (Agilent). mRNA library preparation and RNA sequencing were performed by Novogene. mRNA libraries were prepared with NEBNext Ultra RNA Library Prep Kit for Illumina (NEB) and size selection for libraries was performed using AMPure XP system (Beckman Coulter), followed by library purity analysis. Libraries were sequenced on NovaSeq PE150 and reads mapped to the UCSC mouse reference genome mm10 using STAR. FeatureCounts was used to calculate the expression level as reads per kilobase per million (RPKM). DESeq2 calculated differentially expressed genes with p-values adjusted using Benjamini and Hochberg's method for controlling the false discovery rate (FDR). Genes with significant differential expression were determined by $p < 0.05$. Gene Set Enrichment Analysis was performed with the Molecular Signatures Database, and $-\log_{10}$(p-value) calculated for Hallmark gene sets.

## qPCR

cDNA was synthesized from total RNA using qScript (QuantBio #95048-100). Relative mRNA expression was measured with SsoAdvanced Universal Probes Supermix reactions (Bio-Rad #175284) read out with a QuantStudio 3 real-time PCR system (Applied Biosystems). TATA-box binding protein (*Tbp*) was the invariant control. Roche Universal Probe Gene Expression Assays were used as previously described (*Koh et al., 2018*). The following primer and probe sets were used to detect the following genes: *Ppara*: tcgagttcatgcaagtttcg (F), tccctcctggcttctctagg (R), Roche Universal Probe Library probe #1; *Acox1*: caccattgccattcgataca (F), tgcgtctgaaaatccaaatc (R), Roche Universal Probe Library probe #106; *Ucp1:* ggcctctacgactcagtcca (F), taagccggctgagatcttgt (R), Roche Universal Probe Library probe #34; *Cidea:* aaaccatgaccgaagtagcc (F), aggccagttgtgatgactaagac (R), Roche Universal Probe Library probe #66; *Prdm16:* acaggcaggctaagaaccag (F), cgtggagaggagtgtcttcag (R), Roche Universal Probe Library probe #56; *Pparg2:* gaaagacaacggacaaatcacc (F), gggggtgatatgtttgaacttg (R),

Roche Universal Probe Library probe #7; *Cpt1b:* gagtgactggtgggaagaatatg (F), gctgcttgcacatttgtgtt (R), Roche Universal Probe Library probe #92; *Tbp:* cggtcgcgtcattttctc (F), gggttatcttcacacaccatga (R), and Roche Universal Probe Library probe #51.

## Luciferase assays

We used HepG2 cells purchased directly by our lab for the studies. All cells are tested monthly to ensure an absence of mycoplasma contamination. HepG2 cells (ATCC, HB-8065, RRID:CVCL_002) were maintained in the following media: Eagle's Minimum Essential Media supplemented with 10% FBS, and 1% penicillin/streptomycin antibiotics. We used Lipofectamine 2000 to transiently express pcDNA3.1-PPARA (Addgene #169019), pCMV β-gal (ATCC) and PPRE-luciferase fusion plasmids (Addgene #1015). 20 hr later, cells were treated with lumiflavin (Cayman) or 10 µM WY14643 (Cayman) for an additional 24 hr. 0.1% DMSO served as the vehicle treatment. Reporter gene activity was detected using the Promega Luciferase Assay Kit. Relative luminescence units were normalized to β-galactosidase activity (Sigma).

## Consensome and high-confidence transcriptional (HCT) target intersections

Transcription factor footprint analysis and consensome enrichments were performed (*Ochsner et al., 2019*). For transcription factors, node and node family consensomes are gene lists ranked according to the strength of their regulatory relationship with upstream signaling pathway nodes derived from independent publicly archived transcriptomic or ChIP-Seq datasets. Genes in the 95th percentile of a given node consensome were designated high-confidence transcriptional targets (HCTs) for that node and used as the input for the HCT intersection analysis using the Bioconductor GeneOverlap analysis package implemented in R. For both consensome and HCT intersection analysis, p-values were adjusted for multiple testing using the method of Benjamini and Hochberg to control the FDR as implemented with the p. adjust function in R, to generate q values. Evidence for a transcriptional regulatory relationship between a node and a gene set was inferred from a larger intersection between the gene set and HCTs for a given node or node family than would be expected by chance after FDR correction (q < 0.05). The HCT intersection analysis code has been deposited in the SPP GitHub account at https://github.com/signaling-pathways-project/ominer/ (copy archived at *signaling-pathways-project, 2023*).

## Metabolomics

Targeted measurements of hepatic carnitines, fatty acids, lipids species, CoAs, glycolysis, and TCA metabolites were carried out by the BCM Dan L Duncan Cancer Center CPRIT Cancer Proteomics and Metabolomics Core. Parallel analysis of lipids and ceramides was performed at the University of Utah. All samples were processed and analyzed as described previously (*Chaurasia et al., 2019*; *Kettner et al., 2016*).

### Reagents

High-performance liquid chromatography grade and mass spectrometry grade reagents were used: acetonitrile, methanol, and water (Burdick & Jackson); formic acid, ammonium acetate, and internal standards (Sigma-Aldrich); MS grade lipid standards (Avanti Polar Lipids).

### Internal standards and quality control

To assess overall process reproducibility, mouse pooled liver or serum samples were run along with the experimental samples. A number of internal standards, including injection standards, process standards, and alignment standards, were used to assure QA/QC targets to control for experimental variability. Aliquots (200 µl) of 10 mM solutions of isotopically labeled standards were mixed and diluted up to 8000 µl (final concentration 0.25 mM) and aliquoted into a final volume of 20 µl. The aliquots were dried and stored at –80°C until further analysis. To monitor instrument performance, 20 µl of a matrix-free mixture of the internal standards were reconstituted in 100 µl of methanol:water (50:50) and analyzed by multiple reaction monitoring (MRM). Metabolite extraction from the samples was monitored using pooled mouse serum or liver samples and spiked internal standards. The median coefficient of variation (CV) value for the internal standard compounds was 5%. To address overall

process variability, metabolomic studies included a set of nine experimental sample technical replicates, which were spaced evenly among the injections for each day. The matrix-free internal standards and serum and liver samples were analyzed twice daily.

## Separation of CoAs and carnitines

Targeted profiling for CoAs and carnitines in electrospray ionization-positive mode by the RP chromatographic method employed a gradient containing water (solvent A) and acetonitrile (ACN, solvent B, with both solvents containing 0.1% formic acid). Separation of metabolites was performed on a Zorbax Eclipse XDBC18 column (50 × 4.6 mm i.d.; 1.8 µm, Agilent Technologies) maintained at 37°C. The gradient conditions were 0–6 min in 2% B, 6.5 min in 30% B, 7 min in 90% B, 12 min in 95% B; followed by re-equilibration to the initial conditions.

## Separation of glycolysis and TCA metabolites

Glycolysis and TCA metabolites were separated by normal phase chromatography using solvents containing water (solvent A), solvent A modified by the addition of 5 mM ammonium acetate (pH 9.9), and 100% acetonitrile (solvent B). The binary pump flow rate was 0.2 ml/min with a gradient spanning 80% B to 2% B over 20 min, 2% B to 80% B for 5 min, and 80% B for 13 min. The flow rate was gradually increased during the separation from 0.2 ml/min (0–20 min) to 0.3 ml/min (20–25 min), and then 0.35 ml/min (25–30 min), 0.4 ml/min (30–37.99 min), and finally to 0.2 ml/min (5 min). Metabolites were separated on a Luna Amino (NH2) column (4 µm, 100 A 2.1 × 150 mm, Phenominex) maintained in a temperature-controlled chamber (37°C). All the columns used in this study were washed and reconditioned after every 50 injections.

## Separation of fatty acids

Targeted profiling for fatty acids employed a reverse phase chromatographic method by a gradient containing water (solvent A) with 10 mM ammonium acetate (pH 8) and 100% methanol (solvent B) on a Luna Phyenyl Hexyl column (3 µm, 2 × 150 mm; Phenominex, CA) maintained at 40°C. The binary pump flow rate was 0.2 ml/min with a gradient spanning 40% B to 50% B over 8 min, 50% B to 67% B over 5 min, hold 67% B for 9 min, 67% B to 100% B over 1 min, hold 100% B for 6 min, 100% B to 40% B over 1 min, and hold 40% B for 7 min.

## Liquid chromatography/mass spectrometry (LC/MS) analyses

The chromatographic separation of non-lipid metabolites was performed using either reverse phase separation or normal phase online with the unbiased profiling platform based on a 1290 SL Rapid resolution LC and a 6490 triple Quadrupole mass spectrometer (Agilent Technologies, Santa Clara, CA). Lipidomics required a Shimadzu CTO-20A Nexera X2 UHPLC coupled with TripleTOF 5600 equipped with a Turbo VTM ion source (AB Sciex, Concord, Canada). Using a dual electrospray ionization source, the samples were independently examined in both positive and negative ionization modes. The data acquisition during the analysis was controlled using the Mass Hunter workstation data acquisition software.

## Lipidomics

Mouse liver lipids were extracted using a modified Bligh-Dyer method. Briefly, 50 mg of crushed tissue sample from mouse whole liver was used. A 2:2:2 volume ratio of water/methanol/dichloromethane was used for lipid extract at room temperature after spiking internal standards 17:0 LPC, 17:0 PC, 17:0 PE, 17:0 PG, 17:0 ceramide, 17:0 SM, 17:0 PS, 17:0 PA, 17:0 TAG, 17:0 MAG, DAG 16:0/18:1, CE 17:0. The organic layer was collected, followed by a complete drying procedure under nitrogen. Before MS analysis, the dried extract was resuspended in 100 µl of Buffer B (10:5:85 acetonitrile/water/isopropyl alcohol) containing 10 mM NH$_4$OAc and subjected to reverse-phase chromatography and LC/MS. Internal standards prepared in chloroform/methanol/water (100 pmol/µl) were LPC 17:0/0:0, PG 17:0/17:0, PE 17:0/17:0, PC 17:0/17:0, TAG 17:0/17:0/17:0, SM 18:1/17:0, MAG 17:0, DAG 16:0/18:1, CE 17:0, ceramide 18:1/17:0, PA 17:0, PI 17:0/20:4, and PS 17:0/17:0.

For lipid separation, 5 ml of the lipid extract was injected into a 1.8 mm particle 50 × 2.1 mm Acquity HSS UPLC T3 column (Waters). The column heater temperature was set at 55°C. For chromatographic

elution, a linear gradient was used over a 20 min total run time, with 60% Solvent A (acetonitrile/water [40:60, v/v] with 10 mM ammonium acetate) and 40% Solvent B (acetonitrile/water/isopropanol [10:5:85 v/v] with 10 mM ammonium acetate) gradient in the first 10 min. The gradient was ramped linearly to 100% Solvent B for 7 min. Then the system was switched back to 60% Solvent B and 40% Solvent A for 3 min. A 0.4 ml/min flow rate was used at an injection volume of 5 µl. The column was equilibrated for 3 min and run at a flow rate of 0.4 ml/min for a total run time of 20 min. TOF MS survey scans (150 ms) and 15 MS/MS scans with a total duty cycle time of 2.4 s were performed. The mass range in both modes was 50–1200 m/z. The acquisition of MS/MS spectra by data-dependent acquisition (DDA) function of the Analyst TF software (AB Sciex).

The raw data in .mgf format were converted using ProteoWizard software. The NIST MS PepSearch Program was used to search the converted files against LipidBlast libraries. The m/z width was determined via the mass accuracy of internal standards at 0.001 for positive mode and 0.005 for a negative mode at an overall mass error of less than 2 ppm. The minimum match factor at 400 was set for the downstream data processing. The MS/MS identification results from all the files were combined using an in-house software tool to create a library for quantification. The raw data files were searched against this in-house library of known lipids with mass and retention time using Multiquant 1.1.0.26 (ABsciex). The lipid species identified in the positive or negative ion modes were analyzed separately using relative abundance of peak spectra for the downstream analyses. The identified lipids were quantified by normalizing against their respective internal standard.

## Ceramides and lipids

Lipid extracts are separated on an Acquity UPLC CSH C18 1.7 µm 2.1 × 50 mm column maintained at 60°C connected to an Agilent HiP 1290 Sampler, Agilent 1290 Infinity pump, equipped with an Agilent 1290 Flex Cube and Agilent 6490 triple quadrupole (QQQ) mass spectrometer. In positive ion mode, sphingolipids are detected using dynamic multiple reaction monitoring (dMRM). Source gas temperature is set to 210°C, with an $N_2$ flow of 11 l/min and a nebulizer pressure of 30 psi. Sheath gas temperature is 400°C, sheath gas ($N_2$) flow of 12 l/min, capillary voltage is 4000 V, nozzle voltage 500 V, high-pressure RF 190 V, and low-pressure RF is 120 V. Injection volume is 2 µl, and the samples are analyzed in a randomized order with the pooled QC sample injection eight times throughout the sample queue. Mobile phase A consists of ACN:$H_2O$ (60:40 v/v) in 10 mM ammonium formate and 0.1% formic acid, and mobile phase B consists of IPA:ACN:$H_2O$ (90:9:1 v/v) in 10 mM ammonium formate and 0.1% formic acid. The five chromatography gradient starts at 15% mobile phase B, increases to 30% B over 1 min, increases to 60% B from 1 to 2 min, increases to 80% B from 2 to 10 min, and increases to 99% B from 10 to 10.2 min where it is held until 14 min. Post-time is 5 min, and the flow rate is 0.35 ml/min throughout. Collision energies and cell accelerator voltages were optimized using sphingolipid standards with dMRM transitions as [M+H]+→[m/z = 284.3] for dihydroceramides, [M+H]+→[m/z = 287.3] for isotope-labeled dihydroceramides, [M-$H_2O$+H]+→[m/z = 264.2] for ceramides, [M$H_2O$+H]+→[m/z = 267.2] for isotope-labeled ceramides and [M+H]+→[M-$H_2O$+H] for all targets. Sphingolipids and ceramides without available standards are identified based on HR-LC/MS, quasi-molecular ions, and characteristic product ions. Their retention times are either taken from HR-LC/MS data or inferred from the available standards. Results from LC-MS experiments are collected using Agilent Mass Hunter Workstation and analyzed using the software package Agilent Mass Hunter Quant B.07.00. Ceramide and lipid species are quantitated based on peak area ratios to the standards added to the extracts.

## Analysis of FAD and FMN by IC-HRMS

To determine the relative abundance of FAD and FMN in mouse liver tissue, extracts were prepared and analyzed by high-resolution mass spectrometry (HRMS). Approximately 20–30 mg of tissue were pulverized in liquid nitrogen then homogenized with a Precellys Tissue Homogenizer. Metabolites were extracted using 80/20 (v/v) methanol/water with 0.1% ammonium hydroxide. Samples were centrifuged at 17,000 × g for 5 min at 4°C, supernatants were transferred to clean tubes, followed by evaporation under vacuum. Samples were reconstituted in deionized water, then 10 µl was injected into a Thermo Scientific Dionex ICS-5000+ capillary ion chromatography (IC) system containing a Thermo IonPac AS11 250 × 2 mm 4 µm column. IC flow rate was 360 µl/min (at 30°C), and the gradient conditions are as follows: initial 1 mM KOH, increased to 35 mM at 25 min, increased to 99 mM at

39 min, and held at 99 mM for 10 min. The total run time was 50 min. To increase desolvation for better sensitivity, methanol was delivered by an external pump and combined with the eluent via a low dead volume mixing tee. Data were acquired using a Thermo Orbitrap Fusion Tribrid Mass Spectrometer under ESI negative mode and imported to Thermo Trace Finder software for final analysis. Relative abundance was normalized by tissue weight.

## Analysis of reduced and oxidized coenzymes by triple quadruple LC-MS

To determine the relative abundance of ubiquinone (oxidized CoQ10), ubiquinol (reduced CoQ10), ubiquinone-9 (CoQ9), and ubiquinol-9 (reduced CoQ9) in mouse liver samples, extracts were prepared and analyzed by Thermo Scientific TSQ triple quadrupole mass spectrometer coupled with a Dionex UltiMate 3000 HPLC system. Approximately 20–30 mg of tissue were pulverized in liquid nitrogen then homogenized with a Precellys Tissue Homogenizer. Coenzymes were extracted with 500 µl ice-cold 100% isopropanol. Tissue extracts were vortexed, centrifuged at 17,000 × $g$ for 5 min at 4°C, and supernatants were transferred to clean autosampler vials. The mobile phase was methanol containing 5 mM ammonium formate. Separations of CoQ9, CoQ10, reduced-CoQ9, and reduced-CoQ10 were achieved on a Kinetex 2.6 µm C18 100 Å, 100 × 4.6 mm column. The flow rate was 400 µl/min at 35°C. The mass spectrometer was operated in the MRM positive ion electrospray mode with the following transitions. CoQ10/oxidized: m/z 863.7→197.1 CoQ10/reduced: m/z 882.7→197.1, CoQ9/oxidized: m/z 795.6 →197.1, and CoQ9/reduced: m/z 814.7 →197.1. Raw data files were imported to Thermo Trace Finder software for final analysis. Relative abundance was normalized by tissue weight.

## Quantification and statistical analysis

All measurements were taken from distinct biological samples. Unless otherwise noted, all statistical analyses were performed using GraphPad Prism (version 9) and tests described in the figure legends. In the case of multiple groups, a one- or two-way ANOVA with post-hoc tests were used to determine statistical significance. When only two groups were compared, nonparametric Mann–Whitney tests were used to determine statistical significance. Gene expression and metabolomic heatmaps were plotted as Z-scores using R(4.0.3) and ComplexHeatmap (2.6.2). The species-by-species $t$-test was applied for metabolomics data to identify the top differentially regulated metabolites that passed the nominal threshold p-values. For multiple comparisons, the Sidak procedure was used for false discovery rate (FDR) correction. Statistical analysis of energy balance was performed by ANCOVA with lean body mass as a co-variate (*Mina et al., 2018*). No statistical method was used to predetermine sample size. Unblinded analysis of histology was performed by the investigators. All data are expressed as mean ± SEM, unless otherwise specified.

## Materials availability

This study did not generate new unique reagents. The authors declare that reagents utilized are available upon reasonable request to the corresponding author.

## Acknowledgements

This work was funded by the Nancy Chang, PhD Award for Research Excellence at Baylor College of Medicine, American Diabetes Association #1-18-IBS-105 and NIH R01DK114356 (to SMH). We thank the Gene Vector Core at Baylor College of Medicine for vectors and support. Other BCM core services received support from NCI P30CA125123: Human Tissue Acquisition and Pathology, Metabolomics, and Integrated Microscopy.

## Additional information

### Competing interests

The authors declare that no competing interests exist.

## Funding

| Funder | Grant reference number | Author |
|---|---|---|
| American Diabetes Association | 1-18-IBS-105 | Sean M Hartig |
| National Institute of Diabetes and Digestive and Kidney Diseases | R01DK114356 | Sean M Hartig |

The funders had no role in study design, data collection and interpretation, or the decision to submit the work for publication.

## Author contributions

Peter M Masschelin, Aaron R Cox, Conceptualization, Data curation, Formal analysis, Investigation, Methodology, Writing – original draft, Writing – review and editing; Pradip Saha, Liping Wang, Philip L Lorenzi, Formal analysis, Investigation, Methodology; Scott A Ochsner, Data curation, Software, Formal analysis, Investigation; Kang Ho Kim, Conceptualization, Formal analysis, Investigation, Methodology, Writing – original draft, Writing – review and editing; Jessica B Felix, Formal analysis, Investigation, Writing – review and editing; Robert Sharp, Xin Li, Investigation; Lin Tan, Jun Hyoung Park, Vasanta Putluri, Investigation, Methodology; Alli M Nuotio-Antar, Benny Abraham Kaipparettu, Resources, Investigation, Methodology; Zheng Sun, Resources, Investigation; Nagireddy Putluri, Data curation, Formal analysis, Methodology; David D Moore, Conceptualization, Investigation, Writing – original draft; Scott A Summers, Conceptualization, Resources, Investigation, Methodology; Neil J McKenna, Conceptualization, Resources, Formal analysis, Investigation, Methodology, Writing – review and editing; Sean M Hartig, Conceptualization, Formal analysis, Supervision, Funding acquisition, Investigation, Writing – original draft, Project administration, Writing – review and editing

## Author ORCIDs

Xin Li ⓘ http://orcid.org/0000-0002-7325-4250
Liping Wang ⓘ http://orcid.org/0000-0003-3537-0952
Philip L Lorenzi ⓘ http://orcid.org/0000-0003-0385-7774
Alli M Nuotio-Antar ⓘ http://orcid.org/0000-0002-1830-4868
Zheng Sun ⓘ http://orcid.org/0000-0002-6858-0633
Sean M Hartig ⓘ http://orcid.org/0000-0002-2695-2072

## Ethics

This study was performed in strict accordance with the recommendations in the Guide for the Care and Use of Laboratory Animals of the National Institutes of Health. All of the animals were handled according to approved institutional animal care and use committee (IACUC) protocols (AN-6411) of the Baylor College of Medicine. Every effort was made to minimize suffering.

## Decision letter and Author response

Decision letter https://doi.org/10.7554/eLife.84077.sa1
Author response https://doi.org/10.7554/eLife.84077.sa2

# Additional files

## Supplementary files

• Supplementary file 1. Macronutrient compositions of 90% B2D used in the study.
• Supplementary file 2. Macronutrient compositions of 99% B2D used in the study.
• Supplementary file 3. Antibodies used in this study.
• MDAR checklist

## Data availability

All data generated or analyzed during this study are included in the manuscript and supporting files. RNA-sequencing data have been deposited in GEO under accession code GSE206200. Numerical data used to generate the figures are attached as source data files to each figure. Figure 5 - Source Data 2 and Figure 6 - Source Data 2 contain Western blot scans.

The following dataset was generated:

| Author(s) | Year | Dataset title | Dataset URL | Database and Identifier |
|---|---|---|---|---|
| Masschelin PM, Saha PK, Ochsner SA, Cox AR, Kim K, Felix JB, Sharp R, Li X, Tan L, Park J, Wang L, Putluri V, Lorenzi PL, Sun Z, Kaipparettu B, Putluri N, Moore DD, Summers SA, McKenna NJ, Hartig SM | 2022 | Vitamin B2 enables PPARa regulation of fasting glucose availability | https://www.ncbi.nlm.nih.gov/geo/query/acc.cgi?acc=GSE206200 | NCBI Gene Expression Omnibus, GSE206200 |

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
