## [Editor Report]

This paper provides important findings on the metabolic roles of vitamin B2 in the liver that will be of broad interest. Convincing data establish the effects of vitamin B2 deficiency on body composition, energy expenditure, and glucose metabolism.

---

## [Decision Letter]

**Decision letter after peer review:**

Thank you for submitting your article "Vitamin B2 enables peroxisome proliferator-activated receptor a regulation of fasting glucose availability" for consideration by *eLife*. Your article has been reviewed by 2 peer reviewers, one of whom is a member of our Board of Reviewing Editors, and the evaluation has been overseen by David James as the Senior Editor. The reviewers have opted to remain anonymous.

Essential revisions:

1: The authors state that riboflavin deficiency alters body composition and energy expenditure and that they were surprised the stunted body weight phenotype of B2D did not arise from higher energy expenditure. However, energy expenditure (EE) was not actually calculated to support these two statements. RER should be calculated to reflect glucose homeostasis and metabolic flexibility.

2: The B2D mice likely display lower energy expenditure (estimated from lower O2 consumption), lower activity, and similar food intake, but are leaner. The question is why? What is then the role of FAD for energy balance? Importantly, does B2D induce lipodystrophy in a manner similar to IEMs?

3: I highly recommend the authors subject the mice to metabolic cages during the first 2 weeks of the diet when body weights are still identical between groups, rather than after 4 weeks of the diet. According to PMID:22205519, once obesity or leanness has developed, behavioral and metabolic alterations triggered by the body weight change may obscure or confound the processes that caused the changes in body weight and/or body composition in the first place. Thus, assessing body weight and body composition phenotypes requires a careful analysis of the various factors that might affect these phenotypes. If changes in caloric intake or energy expenditure are observed before the change in body weight or fat mass, it is more likely that one can attribute the difference to these changes.

5. As mammals cannot synthesize vitamin B2 and diet remains the only available source of FAD, why reducing vitamin B2 in the diet by 90% does not impact liver FAD levels nor influence energy balance (Figure S1)? How do the mice adapt to a 90% reduction in vitamin B2 intake? Is a 90% reduction enough to monitor the complete metabolic phenotype of vitamin B12 deficiency?

6. The authors state that FAD is required for hepatic glucose production during fasting. However, the data presented in Figure 2a show that liver FAD displays the opposite 24-h rhythm to liver glucose. It is not ideal to show hepatic glucose level, as the liver secrete glucose into the circulation. The authors should instead plot the 24-h rhythm of fed/fasted liver FAD levels and the corresponding fed/fasted blood glucose levels. According to previous publications, liver FAD levels usually coincide with blood glucose.

7. Authors normalized relative abundances/fold changes from metabolomics measurements to tissue weight. Normalization to the total protein level would be more suitable.

8. In Figure 4b, it is surprising to observe that the body weights of the mice are significantly different at week 0 before any diet interventions. The authors should provide an explanation for this data. This is in fact different from the data presented in Figure 1d (data shown in % for some reason). Another question is why this experiment used 9 weeks of diet while other experiments used 4 weeks of diet. Figure 1d, and Figure 3d showed % weight change during the 4 weeks of diet, and absolute body weights should also be shown during the 4 weeks as shown in Figure 4b. Consistency in presenting the data would help in comparing the results of different experiments.

9. Figure 5d shows a representative protein blotted for each complex (Total OXPHOS Rodent WB Antibody Cocktail, ab110413), which unfortunately does not represent the whole complex. Thus, the authors cannot conclude that B2D and fenofibrate lacked meaningful impacts on relative levels of oxidative phosphorylation proteins. It would be surprising that, if PPARa activity is impacted, some genes encoding various subunits of these complexes would not change as many are known as direct targets of PPARa.

10. For the heatmaps shown in Figure 6a, 6d, the authors compared Ctrl vs B2D, and Ctrl FF vs B2D FF individually, with no comparison between -FF and +FF groups because they are not comparable considering the different lengths of diet (4 weeks vs. 9 weeks) and no placebo gavage for the -FF group. Nonetheless, the authors compared 4 groups together in Figure 5a, 5b, 5c, 5d, 5e, and 6e. Are they different samples? If not, this analysis is flawed.

11. In Figure 2b, the authors perform an "HGP" (hepatic glucose production). However, in the methods, they state that they inject radioactive glucose, and then measure plasma glucose. This experiment would test glucose consumption and not production. Next to this panel, there is a pyruvate tolerance test which is the correct experiment. Is (a) the HGP experimental methods a typo and they actually used radioactive pyruvate? Or did they (b) actually measure glucose consumption by injecting radioactive glucose? If (a), please correct this typo. If (b), test glucose production by injecting radioactive pyruvate rather than radioactive glucose if possible.

---

## [Author Response]

Essential revisions:1: The authors state that riboflavin deficiency alters body composition and energy expenditure and that they were surprised the stunted body weight phenotype of B2D did not arise from higher energy expenditure. However, energy expenditure (EE) was not actually calculated to support these two statements. RER should be calculated to reflect glucose homeostasis and metabolic flexibility.

Thank you for the suggestion. We included measurements of energy expenditure and RER in the revised manuscript, which support our conclusion that increased energy expenditure does not account for the weight differences between B2D and control. Please see revised Figures 1, 3, and Supplemental Figures 1 and 2.

2: The B2D mice likely display lower energy expenditure (estimated from lower O2 consumption), lower activity, and similar food intake, but are leaner. The question is why? What is then the role of FAD for energy balance? Importantly, does B2D induce lipodystrophy in a manner similar to IEMs?

Our findings show that B2D lowers energy expenditure and reduces body weight but we were unable to detect changes in food intake.

To our knowledge, almost no studies explored the requirements for FAD in energy balance in a comprehensive, mechanistic way. This lack of attention to a critical electron transfer cofactor and regulator of diverse flavoproteins motivated our study. We also searched extensively for energy expenditure analysis of animal models and people of IEMs resembling B2D. Only a handful of studies addressed the issue. IEMs that arise from mutations in genes encoding mitochondrial FAD transfer enzymes for FAO are often identified by ectopic fat distribution and liver disease (J Inherit Metab Dis 42:608–619, 2019). People with long-chain fatty acid oxidation disorders (LC-FAODs), including mutations in the FAD-dependent protein ACADVL, show similar energy expenditure phenotypes as B2D. Energy expenditure is significantly lower among subjects with LC-FAODs compared to the reference population in the National Health and Nutrition Examination Survey (Mol Genet Metab 138:107519, 2023). These observations suggest responses to B2D are relevant to studying IEMs defined by mutations in FAD-dependent metabolic enzymes.

In new efforts to explore energy balance in this model, we examined markers of thermogenesis. Control diet and B2D showed similar brown adipose tissue (BAT) morphology and gene expression patterns (Supplemental Figure 3). Overall our results show that B2D causes an energy conservation phenotype with reduced energy expenditure. It remains unclear whether this is a primary or compensatory change in response to B2D. We have clarified this point in the discussion.

3: I highly recommend the authors subject the mice to metabolic cages during the first 2 weeks of the diet when body weights are still identical between groups, rather than after 4 weeks of the diet. According to PMID:22205519, once obesity or leanness has developed, behavioral and metabolic alterations triggered by the body weight change may obscure or confound the processes that caused the changes in body weight and/or body composition in the first place. Thus, assessing body weight and body composition phenotypes requires a careful analysis of the various factors that might affect these phenotypes. If changes in caloric intake or energy expenditure are observed before the change in body weight or fat mass, it is more likely that one can attribute the difference to these changes.

We agree that body weight changes represent complex outcomes that require careful analysis. In our studies, we follow the most contemporary analysis protocols, including the use of CalR (Cell Metab 2018, 28:656-666) to determine whether energy expenditure variables change as a result of the diet. For this phenotype, body mass variables (without normalization to lean body mass) were accounted for in our statistical analyses to report the energy expenditure data accurately. We cannot find other reported mouse models of IEM where energy expenditure was analyzed by CalR to identify body-weight independent effects. As we now discuss in the manuscript (revised Discussion section), we note that B2D effects resemble energy expenditure measurements for people with IEM (Mol Genet Metab 138:107519, 2023). To further address the reviewer's concern, we performed measurements of food intake during the first week of B2D (Author response image 1). One week of B2D did not impact food intake. We know additional experiments and equipment time will be needed to address the complex energy balance effects of B2D, and this will be the focus of our future work.

**Author response image 1. sa2fig1:** One-week of B2D does not affect food intake or ad libitum blood glucose. Individually housed one-month old male mice (n=3 per group) were exposed to 99% B2D or isocaloric control diet (Ctrl) for one week. (a) Cumulative food intake. (b) ad libitum blood glucose. Data are represented as mean +/- SEM. Mann-Whitney test for a and b.

4. As mammals cannot synthesize vitamin B2 and diet remains the only available source of FAD, why reducing vitamin B2 in the diet by 90% does not impact liver FAD levels nor influence energy balance (Figure S1)? How do the mice adapt to a 90% reduction in vitamin B2 intake? Is a 90% reduction enough to monitor the complete metabolic phenotype of vitamin B12 deficiency?

Our results show that three weeks of 90% B2D did not influence whole-body metabolic phenotypes. Based on data reported by Fenton and Cowgill (J Nutr 34:273–283, 1947), mice require about 4 mg riboflavin/kg diet for normal growth. This level varies among mouse strains. However, mouse chow generally contains 6 or 7 mg riboflavin/kg diet (American Institute of Nutrition, 1977). Most tissues store very little of the vitamin as riboflavin. The majority of bioavailable riboflavin is converted to FAD and excess is excreted in urine as FMN (N Engl J Med 283:463-472, 1970). Riboflavin turnover rates in rodents are also relatively slow at approximately 16 days (J Nutr 93:445-453, 1967), which underscores sufficient FAD concentrations remain in the liver after four weeks of 90% B2D. Therefore, the historical literature suggests 10% of the required level of riboflavin may be sufficient for growth and metabolism in the mouse in our four week monitoring period. In our revision, we expanded our comments on the riboflavin requirements to support mouse body weight gain and placed our observations within the nutritional studies performed almost 80 years ago. We are unaware whether the riboflavin requirements for C57BL/6 mice have ever been revisited.

5. The authors state that FAD is required for hepatic glucose production during fasting. However, the data presented in Figure 2a show that liver FAD displays the opposite 24-h rhythm to liver glucose. It is not ideal to show hepatic glucose level, as the liver secrete glucose into the circulation. The authors should instead plot the 24-h rhythm of fed/fasted liver FAD levels and the corresponding fed/fasted blood glucose levels. According to previous publications, liver FAD levels usually coincide with blood glucose.

Thank you for the excellent suggestion. Revised Figure 2 includes liver FAD levels in the fasted and re-fed state, which demonstrates liver FAD levels coincide with blood glucose levels.

6. Authors normalized relative abundances/fold changes from metabolomics measurements to tissue weight. Normalization to the total protein level would be more suitable.

We have extensively compared metabolomic data normalized to total DNA, total protein, tissue weight, and total signal, and we’ve found that total DNA, tissue weight, and total signal all perform equally well and better than total protein. The problem with total protein is that it cannot be measured accurately after metabolite extraction, whereas total DNA, tissue weight, and total signal can all be measured accurately after metabolite extraction. Our previous publication (Anal Chem 85:9536-9542, 2013) laid the groundwork for accurate metabolomic data normalization, and over the past ten years since that publication we’ve optimized protocols for accurate measurement of tissue weight and found the approach yields reproducible results.

7. In Figure 4b, it is surprising to observe that the body weights of the mice are significantly different at week 0 before any diet interventions. The authors should provide an explanation for this data. This is in fact different from the data presented in Figure 1d (data shown in % for some reason). Another question is why this experiment used 9 weeks of diet while other experiments used 4 weeks of diet. Figure 1d, and Figure 3d showed % weight change during the 4 weeks of diet, and absolute body weights should also be shown during the 4 weeks as shown in Figure 4b. Consistency in presenting the data would help in comparing the results of different experiments.

Thank you for pointing this out. Body weight % measurements equalize the average weight of the two groups before treatment, enabling a clear graphical depiction of the change elicited by the interventions from the same starting point. Nonetheless, we acknowledge the previous representation of body weight change may cause confusion. Animals were randomly assigned to control or B2D and we did not attempt to match initial body weights, therefore any differences in initial body weight occurred by chance. In the process of re-assessing the statistics for Figure 4D, we do not find body weights are different at week 0 or week 1. To that end, we revised all figures to show body weight (g) and body weight gain (%) for all experiments to facilitate comparison of the results of the different experiments.

Regarding the experimental design, we exposed mice to B2D long enough to durably establish steady-state weight gain effects before fenofibrate exposure. We used malnutrition (J Hepatol 65:1198-1208, 2016), Ppara knockout (Gut 65:1202-1214, 2016), cancer cachexia (Proc Natl Acad Sci USA 115: E743–E752, 2018), and other riboflavin-depletion studies (Ann Neurol 84:659-673, 2018) as guides to design the B2D experiments and fenofibrate interventions. The revised methods section cites these studies.

8. Figure 5d shows a representative protein blotted for each complex (Total OXPHOS Rodent WB Antibody Cocktail, ab110413), which unfortunately does not represent the whole complex. Thus, the authors cannot conclude that B2D and fenofibrate lacked meaningful impacts on relative levels of oxidative phosphorylation proteins. It would be surprising that, if PPARa activity is impacted, some genes encoding various subunits of these complexes would not change as many are known as direct targets of PPARa.

We agree and edited the text accordingly to note the OXPHOS antibody only detects a fraction of mitochondrial proteins. In the discussion, we noted that fibrates show a minor impact on OXPHOS complexes only at high concentrations compared to other lipid and glucose lowering agents, such as statins and thiazolidinediones (Toxicol Appl Pharmacol 223: 277-287, 2007).

9. For the heatmaps shown in Figure 6a, 6d, the authors compared Ctrl vs B2D, and Ctrl FF vs B2D FF individually, with no comparison between -FF and +FF groups because they are not comparable considering the different lengths of diet (4 weeks vs. 9 weeks) and no placebo gavage for the -FF group. Nonetheless, the authors compared 4 groups together in Figure 5a, 5b, 5c, 5d, 5e, and 6e. Are they different samples? If not, this analysis is flawed.

We appreciate your input. Although the diet treatments are different, we feel inclusion of these contrasts remains important for the study. The fenofibrate treatments uncover new gene and metabolic programs beyond B2D alone, including greater activation of the integrated stress response. The text and figure legends have been revised to clarify sampling times for these experiments. To perform new experiments and associated mass spectrometry analysis, we would need to repeat experiments on four additional groups of mice and home in on appropriate times for sampling. This is non-trivial and beyond the scope of the current project.

10. In Figure 2b, the authors perform an "HGP" (hepatic glucose production). However, in the methods, they state that they inject radioactive glucose, and then measure plasma glucose. This experiment would test glucose consumption and not production. Next to this panel, there is a pyruvate tolerance test which is the correct experiment. Is (a) the HGP experimental methods a typo and they actually used radioactive pyruvate? Or did they (b) actually measure glucose consumption by injecting radioactive glucose? If (a), please correct this typo. If (b), test glucose production by injecting radioactive pyruvate rather than radioactive glucose if possible.

We clarified the description of the glucose metabolism data in Figure 2 and separated the pyruvate tolerance test and the hepatic glucose production panels. Our lab performed studies of infused (not injected) radiolabeled glucose to assess basal hepatic glucose production at steady state. The revised Materials and methods includes additional details of the protocol. Additional details regarding the protocol can be found at the MMPC: https://www.mmpc.org/shared/document.aspx?id=137&docType=Protocol.